# A structural code for assembly specificity in GID/CTLH-type E3 ligases

Pia Maria van gen Hassend, Hermann Schindelin*

Institute of Structural Biology, Rudolf Virchow Center for Integrative and Translational Bioimaging, Julius-Maximilians-Universität Würzburg, Würzburg, Germany

## eLife Assessment

This structural biology study provides insights into the assembly of the GID/CTLH E3 ligase complex. The multi-subunit complex forms unique, ring-shaped assemblies and the findings presented here describe a "specificity code" that regulates formation of subunit interfaces. The data supporting the conclusions are **convincing**, both in thoroughness and rigor. This study will be **valuable** to biochemists, structural biologists, and could lay foundation for novel designed protein assemblies.

*For correspondence:
hermann.schindelin@virchow.uni-wuerzburg.de

Competing interest: The authors declare that no competing interests exist.

**Abstract** GID/CTLH-type E3 ligases assemble into conserved ring-shaped architectures built from repeating LisH-CTLH-CRA modules, yet the molecular rules that enforce their highly specific subunit arrangement have remained unknown. Here, we decode the structural 'assembly specificity code' that governs CRA-CRA pairing. Using crystal structures of multiple CTLH-CRA domains, including the RanBP9-muskelin heterodimer, integrated with quantitative binding analyses, we show that several interfaces operate with exceptionally high affinity, reaching the picomolar range, and that conserved sequence and geometric features enable each subunit to only select cognate partners. Strikingly, targeted perturbations of these features are sufficient to reprogram pairing preferences, enabling engineered subunits such as RanBP10 or Twa1 to adopt non-native interaction partners. These findings reveal the molecular logic that preserves the architecture of GID/CTLH-type E3 ligases and demonstrate that their assembly code is both decipherable and engineerable, providing a conceptual foundation for reconfiguring these ring-shaped E3 ligases.

## Introduction

Ubiquitylation – the covalent attachment of the small protein ubiquitin to target substrates – is a central post-translational modification in eukaryotes (*Hershko and Ciechanover, 1998*). Depending on chain type, length, and topology, ubiquitylation can signal for proteasomal degradation, alter subcellular localization, or regulate protein interactions (*Komander and Rape, 2012*). Specificity is conferred by E3 ligases, which recruit ubiquitin-loaded E2 enzymes (*Zheng and Shabek, 2017*). Most E3s belong to the RING family, where the RING domain directly binds the E2~ubiquitin conjugate ('~' denotes a thioester link between a cysteine in the E2 and the C-terminal glycine of ubiquitin), while separate regions of the ligase mediate substrate recognition (*Deshaies and Joazeiro, 2009*).

The GID/CTLH-type RING ligases form a conserved family of multi-subunit assemblies that share a common architectural framework but vary in composition across species. In yeast, this ligase is known as the GID (glucose-induced-deficient) complex, whereas in mammals, it is referred to as the CTLH (C-terminal to LisH) complex; functionally, both represent variations of the same underlying E3 platform (*Alpi et al., 2025*).

A defining feature of this ligase family is its use of the dedicated E2 enzyme Ube2H, which exemplifies a remarkably specific and evolutionarily conserved E2-E3 pairing (*Chen et al., 2017*; *Chrustowicz*

*et al., 2024*; *Regelmann et al., 2003*; *Sherpa et al., 2022*; *Zavortink et al., 2020*). Ube2H engages the RING domain through the canonical E2-E3 interface but is further stabilized by an additional phosphorylation-dependent interface (*Chrustowicz et al., 2024*).

Even more exceptional is the structural organization of the CTLH complex. Each core subunit contains a LisH (lissencephaly-1 homology) motif (*Emes and Ponting, 2001*), an eponymous CTLH (*Kobayashi et al., 2007*) motif, and a CRA (CT-11-RanBPM) (*Menon et al., 2004*) motif (*Figure 1A*). The CRA motif is bipartite: the N-terminal segment (CRA$^N$) folds together with the CTLH motif to form a helical CTLH-CRA$^N$ domain, whereas the C-terminal helix (CRA$^C$) bends toward the LisH motif and stabilizes the LisH-mediated dimer through additional interactions (*Qiao et al., 2020*; *Figure 1B*). Through alternating dimers of LisH-CRA$^C$ and CTLH-CRA$^N$ domains, the complex assembles into a characteristic ring-shaped scaffold (*Figure 1C*) of exceptional size (>1 MDa) (*Chrustowicz et al., 2024*; *Mohamed et al., 2021*; *Sherpa et al., 2021*).

Outside the CTLH complex, the LisH-CTLH-CRA architecture occurs only sporadically and has been structurally confirmed in a few cases: the human splicing protein SMU1 (*Ashraf et al., 2019*), the mouse microtubule-associated protein Wdr47 (*Ren et al., 2022*), and the plant transcriptional co-repressors TOPLESS and TOPLESS-related protein 2 (TPR2) (*Ke et al., 2015*; *Ma et al., 2017*; *Martin-Arevalillo et al., 2017*). In these proteins, only the LisH and CTLH motifs are sequence-annotated, while the CRA-like region is inferred from the shared fold. As this architecture is rare elsewhere, the high density of LisH-CTLH-CRA modules within the CTLH complex is especially striking. To date, no other multi-subunit complex – E3 ligase or otherwise – has been shown to use this architecture to assemble.

Comparative genomics show that versions of the CTLH complex core proteins (Rmnd5, Maea, Twa1, RanBP9, and Wdr26) were already present in the last eukaryotic common ancestor, implying that the CTLH scaffold emerged early in eukaryotic evolution (*Francis et al., 2013*; *Tomaštíková et al., 2012*). Subsequent duplications diversified the complex further and allowed alternative assemblies – e.g., Rmnd5 split into A and B paralogs, and RanBP9 duplicated to RanBP10. By contrast, muskelin appears metazoan-specific, representing a later innovation that expanded function without altering the underlying scaffold (*Francis et al., 2013*).

Overall, the ring-shaped scaffold comprises eight LisH-CRA$^C$ interactions interlinked by eight CTLH-CRA$^N$ contacts (*Figure 1C*). The three distinct LisH-CRA$^C$ dimers are: (i) the RING pair Rmnd5a-Maea (further stabilized by their coiled-coil segments; *Chrustowicz et al., 2024*), (ii) RanBP9/10-Twa1, and (iii) homodimers of Wdr26 or muskelin (note: muskelin dimerizes via LisH only [*Barbulescu et al., 2024*; *Delto et al., 2015*], lacking a CRA$^C$ helix). The two CTLH-CRA$^N$ links are: (i) Twa1 with Maea or Rmnd5a and (ii) RanBP9/10 with Wdr26 or muskelin. Together, two Rmnd5a-Maea dimers connect via four RanBP9/10-Twa1 dimers to two Wdr26 or muskelin homodimers, forming a platform for substrates and receptors positioned at the ring's center (*Figure 1A*; *Qiao et al., 2020*; *Sherpa et al., 2021*; *Mohamed et al., 2021*; *Lampert et al., 2018*; *van gen Hassend et al., 2023*; *Gottemukkala et al., 2024*).

Substrates bind either directly to the core (e.g. through muskelin or Wdr26) or via dedicated receptors (such as Gid4, Ypel5 [*Gottemukkala et al., 2024*], or FAM72A [*Barbulescu et al., 2024*; *Feng et al., 2025*]), or a combination of both. Gid4 tethers to adaptor protein Armc8, which in turn is anchored by Twa1 (*Qiao et al., 2020*; *Sherpa et al., 2022*; *van gen Hassend et al., 2023*); Ypel5 binds the WD40 domain of Wdr26, and FAM72A engages muskelin's discoidin and kelch domains (*Barbulescu et al., 2024*; *Figure 1C*). In essence, a modular, variable core combined with interchangeable substrate receptors creates a combinatorial platform that broadens substrate recognition across diverse pathways, including metabolic regulation (*Gottemukkala et al., 2024*; *Leal-Esteban et al., 2018*; *Pham et al., 2025*; *Yi et al., 2024*), modulation of bacterial growth (*Simwela et al., 2024*), cancer development (*Amin et al., 2015*; *Fan et al., 2014*; *Huffman et al., 2019*; *Jiang et al., 2015*; *Orlacchio et al., 2025*; *Sun et al., 2021*), aging (*Liu et al., 2020*), DNA repair (*Barbulescu et al., 2024*; *Palmieri et al., 2018*), erythroid maturation (*Soni et al., 2008*; *Zhen et al., 2020*; *Sherpa et al., 2022*), B-cell development (*Barbulescu et al., 2025*), and other developmental processes such as early embryogenesis (maternal-to-zygotic transition [*Briney et al., 2025*; *Cao et al., 2020*; *Zavortink et al., 2020*]) and neurodevelopment (*Gross et al., 2024*; *Hantel et al., 2022*; *Lu et al., 2017*; *Palavicini et al., 2013*; *Pfirrmann et al., 2015*; *Skraban et al., 2017*; *Zeinali et al., 2025*).

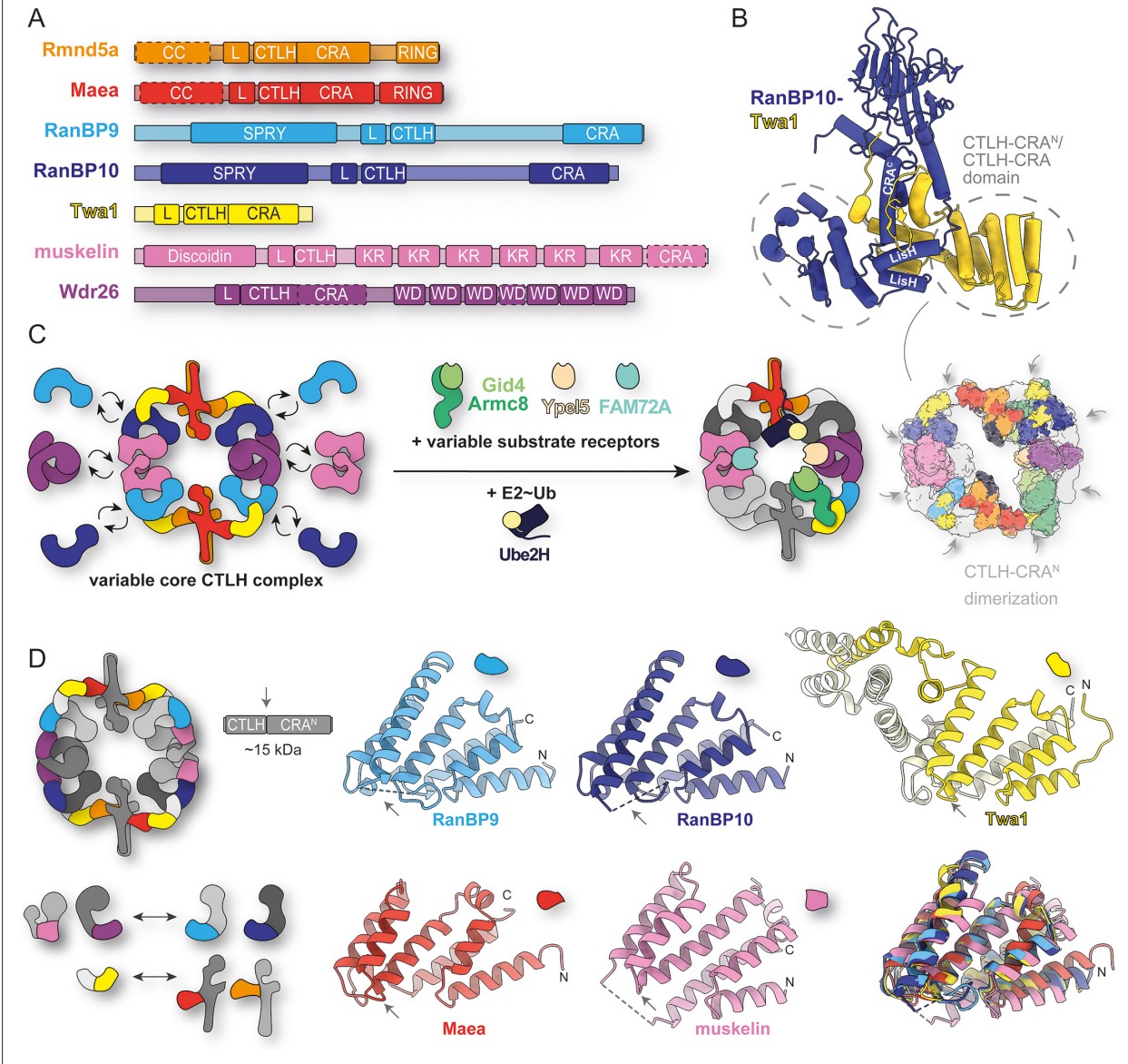

**Figure 1.** Shared structural architecture of CTLH-CRA domains. (**A**) Domain architecture of the core subunits containing LisH (L), CTLH, and CRA motifs. (**B**) Crystal structure of the RanBP10-Twa1 complex (3.2 Å, PDB: 9SOC). LisH and CRA$^C$ helices are labeled. (**C**) Schematic representation of the modular CTLH core complex highlighting interchangeable subunits (RanBP9 ↔ RanBP10; muskelin ↔ Wdr26) and variable substrate receptors (Gid4-Armc8, Ypel5, and FAM72A), illustrating the combinatorial nature of subunit arrangements. Gray arrows mark CTLH-CRA$^N$/CTLH-CRA domain heterodimerization interfaces in a composite model of known structures (PDB: 8PJN, 7NSC, 8QBN, 8TTQ, and newly solved 9SOC) fitted into available cryo-EM maps (EMDB: 12542, 12545, 12547, 17713, 17715, 18172, 45088, 45138, and 45186). (**D**) Schematic overview of CTLH-CRA$^N$/CTLH-CRA domain fusion constructs showing their position and binding specificity within the CTLH core complex. Crystal structures of RanBP9, RanBP10, Twa1, Maea, and muskelin (from RanBP9-muskelin complex) reveal a conserved fold of the CTLH-CRA domain.

The online version of this article includes the following figure supplement(s) for figure 1:

**Figure supplement 1.** Conservation of mammalian core CTLH complex subunits.

**Figure supplement 2.** Comparison of LisH-CRA$^C$ interfaces.

**Figure supplement 3.** Biochemical characterization of different CTLH-CRA domain complexes.

So far, only a limited set of substrates has been mapped to their recognition route, and the repertoire diverges between species. In yeast, Gid4 recognizes gluconeogenic enzymes (Fbp1, Icl1, Mdh2, Pck1) via their N-terminal proline (Pro/N-end rule) (*Chen et al., 2021*; *Chen et al., 2017*; *Hämmerle et al., 1998*; *Regelmann et al., 2003*). In mammals, the CTLH complex does not target these enzymes;

instead, it ubiquitylates the Gid4-dependent receptor-signaling protein ZMYND19 (*Mohamed et al., 2021*) and the Rho-GTPase activator ARHGAP11A (*Bagci et al., 2024*) but without an N-terminal Pro requirement. Additional examples include the base excision repair protein UNG2 (*Barbulescu et al., 2024*) via FAM72A, the metabolic protein NMNAT1 (*Gottemukkala et al., 2024*) and the transcriptional regulator HBP1 (*Lampert et al., 2018*; *Mohamed et al., 2021*) via Wdr26, and the mevalonate pathway protein HMGCS1 (*Maitland et al., 2024*; *Owens et al., 2024*; *Yi et al., 2024*) via muskelin and Gid4 (Pro-dependent).

Despite substantial progress, high-resolution structural coverage of the mammalian CTLH complex and its substrates remains limited. Existing cryo-EM reconstructions define the overall architecture but lack atomic detail for several regions – most notably the CTLH-CRA$^N$ domains (*Figure 1C*). Consequently, key aspects of assembly and specificity have remained unresolved. Moreover, most available structures contain RanBP9; incorporation of RanBP10 is presumed to be similar based on a low-resolution cryo-EM map (*Sherpa et al., 2022*), but has not yet been characterized in detail.

To close this gap, we determined crystal structures of several mammalian CTLH complex components: the RanBP10-Twa1 heterodimer, and the CTLH-CRA$^N$ domains of RanBP9, RanBP10, Maea, Twa1, and the RanBP9-muskelin complex. Guided by these structures, we combined targeted mutagenesis with quantitative binding assays to map residues governing partner specificity and to reprogram pairing preferences. We engineered RanBP10 variants that switch binding preference from muskelin to Maea, and Twa1 variants that switch from Maea to muskelin. We also determined structures of a RanBP10 specificity-switch mutant alone and in complex with Maea, as well as a Twa1 mutant bound to Maea. Together, these results define a CTLH-CRA$^N$ specificity code, illustrating the evolutionary diversification of this unique type of E3 ligase scaffold. This engineering toolkit might be used in future studies for the controlled assembly of alternative ring compositions, thus enabling analysis of substrate dependencies or the design entirely new protein ring assemblies.

## Results

### Structural basis of LisH-CRA$^C$-driven RanBP10-Twa1 heterodimerization

To complement existing cryo-EM data, we pursued X-ray crystallography on the mammalian CTLH complex. All subunits were murine except muskelin, which was from rat (established lab protocols [*Delto et al., 2015*]; 99.6% identical to mouse). Overall, mouse/rat protein versions are highly similar to the human proteins, and the interfaces are well conserved. Differences arise mostly in flexible linker regions or flexible termini (*Figure 1—figure supplement 1*). Due to constraints in soluble expression (*van gen Hassend et al., 2023*) and crystallization, only the near-full-length RanBP10-Twa1 heterodimer yielded diffracting crystals (3.2 Å, highly anisotropic; PDB 9SOC; *Figure 1B*).

As expected, the RanBP10-Twa1 heterodimer adopts the canonical helix arrangement reported for the RanBP9-Twa1 complex (PDB: 7NSC; *Sherpa et al., 2021*), explaining how RanBP10 can substitute for RanBP9 in complex assembly (*Sherpa et al., 2022*). Inspection of their LisH-CRA$^C$-interfaces – dominated by hydrophobic interactions – confirms the structural similarity (*Figure 1—figure supplement 2*). Comparison with other LisH-mediated dimers shows that TBL1 and Lis1 also carry a CRA$^C$-like helix despite lacking a CRA$^N$ segment, underscoring the functional separation of CRA$^N$ and CRA$^C$ motifs.

Across LisH-mediated dimers, conserved leucines and phenylalanines form the hydrophobic core of the two paired helices. Specificity likely arises from partner-specific contacts; e.g., RanBP9/10 (Y382/270) can form a hydrogen bond with Twa1 (R26), whereas Twa1 itself cannot (F42). This difference may explain why RanBP9/10 outcompetes Twa1 homodimerization. Direct mutational tests of this hypothesis, however, were not feasible, as RanBP9/10 express insolubly in *Escherichia coli*, probably due to their exposed hydrophobic LisH-CRA$^C$ interface (*van gen Hassend et al., 2023*).

### Shared structural architecture of CTLH-CRA domains

To obtain higher-resolution data, we designed smaller RanBP9/10 constructs. We omitted the hydrophobic LisH-CRA$^C$ interface and the adjacent SPRY domain, removed the flexible linker between the CTLH and CRA segments (*Figures 1*), and generated fusions of the CTLH and CRA motifs with and without its CRA$^C$ segment (*Figure 1D*). This region was the only part of RanBP9 unresolved in earlier cryo-EM work (PDB 7NSC, *Figure 1C*).

We determined a 1.8 Å crystal structure of the RanBP9 CTLH-CRA construct (PDB: 9SNE). Although the construct included the CRA$^C$ helix, it was not resolved, reinforcing that this helix behaves as an extension of the LisH domain rather than as part of the CTLH-CRA domain. Therefore, and for simplicity, the term 'CTLH-CRA domain' below refers to the functionally relevant CTLH-CRA$^N$ unit.

We next solved CTLH-CRA structures of RanBP10 (2.1 Å, PDB: 9SNF), Twa1 (2.1 Å, PDB: 9SNI), and Maea (1.4 Å, PDB: 9SNH) (*Figure 1D*). CTLH-CRA domains of Wdr26 and Rmnd5a could not be purified individually but were co-expressed with their binding partners (*Figure 1—figure supplement 3A and B*). As with RanBP9/10, removing hydrophobic interfaces enabled soluble expression of the otherwise insoluble Maea and Rmnd5a (in *E. coli*).

Although helix swapping was observed in the Twa1 CTLH-CRA crystal structure (*Figures 1*), where the CRA segment engages a neighboring CTLH motif instead of folding back onto its own, this arrangement most likely arises from crystal packing. The swapped conformation creates an additional dimerization interface that is stabilized within the crystal lattice and was therefore preferentially selected during crystallization. In the context of the full-length protein, such helix swapping is unlikely to occur, as dimerization mediated by the CRA$^C$ helix would restrain the CRA segment (*Figure 1B*).

Muskelin lacks a CRA$^C$ helix, and its C-terminus (Ct) functions as a CRA$^N$-like module. The muskelin CTLH-CRA domain fusion (previously termed CTLH-Ct construct [*van gen Hassend et al., 2023*]) was soluble on its own but did not yield crystals. Removing a few amino acids at the C-terminus rendered it insoluble, which could be rescued by co-expression with RanBP9/10 CTLH-CRA domains (*Figure 1—figure supplement 3A*), thus allowing us to determine the crystal structure of the co-expressed muskelin-RanBP9 CTLH-CRA domain complex (2.0 Å, PDB: 9SNV) (*Figure 2A*).

Overall, all CTLH-CRA domains share a similar fold, with four helices contributed by the CTLH motif and four by the CRA motif. Superposition of the experimentally solved CTLH-CRA domain structures with full-length AlphaFold predictions (*Varadi et al., 2022*; *Varadi et al., 2024*) confirmed that truncations and fusion approaches did not affect the overall fold or the canonical dimerization interface (*Figure 1—figure supplement 1*), validating the domain constructs for interface characterization.

## Structural determinants of RanBP9/10-muskelin CTLH-CRA specificity

The assembly interface is mediated by the first two helices of the CRA segment (*Figure 2A*). Sequence alignment of this region (*Figure 2B*) revealed several conserved residues, most notably a leucine near the end of the second CRA helix (marked with * for reference). To pinpoint residues defining RanBP9/10-muskelin pairing, we overlaid the RanBP9-muskelin structure with CTLH-CRA domains of RanBP10 (which binds muskelin) and Twa1 (which does not). For Twa1, we used the lower-resolution RanBP10-Twa1 heterodimer structure (*Figure 2C*), as the crystal structure of the isolated Twa1 CTLH-CRA domain exhibited helix swapping (*Figure 1D*).

In the RanBP9-muskelin structure (*Figure 2C*), muskelin interacts with RanBP9 through two specific contacts: (i) π-π stacking between muskelin F686 and RanBP9 F576 (RanBP10 F543) and (ii) a hydrogen bond between muskelin Q679 and RanBP9 Y581 (RanBP10 Y548). In contrast, Twa1 carries a leucine (L155) and a phenylalanine (F152) at the equivalent positions, preventing these interactions. Moreover, Twa1 contains an additional phenylalanine (F160) at a site where RanBP9/10 features a valine (V589/V556), thus creating steric hindrance with muskelin Q679.

To test these predictions, we generated RanBP10 CTLH-CRA domain mutants (chosen over RanBP9 for better protein handling) with Twa1-like substitutions: F543L, Y548F, and V556F. We constructed single (L--, -F-, and --F), double (LF-, -FF, and L-F), and triple (LFF) mutants and analyzed their binding to muskelin using native agarose gel electrophoresis (NAGE; *Figure 1—figure supplement 3C*) and isothermal titration calorimetry (ITC; *Figure 2*, *Figure 2—figure supplement 1*). Attempts to generate muskelin mutants were largely unsuccessful, with only the F686L variant being purifiable. Wild-type muskelin bound RanBP10 tightly, with a $K_D$ of 4.33±0.99 nM. Single mutations reduced affinity to the double-digit nanomolar range, double mutants into the triple-digit range, and the triple mutant abolished binding – underscoring the importance of these specific contacts and explaining why Twa1 cannot bind muskelin.

To explore why RanBP9/10 cannot bind Maea, we compared the RanBP9-muskelin interface with our Maea CTLH-CRA structure (*Figure 2E*). The overlay revealed a steric clash between a glutamine in RanBP9/10 (Q552/519) and a phenylalanine in Maea (F245). At the equivalent position, muskelin carries a sterically less demanding lysine residue, since its phenylalanine (F686, responsible for π-π

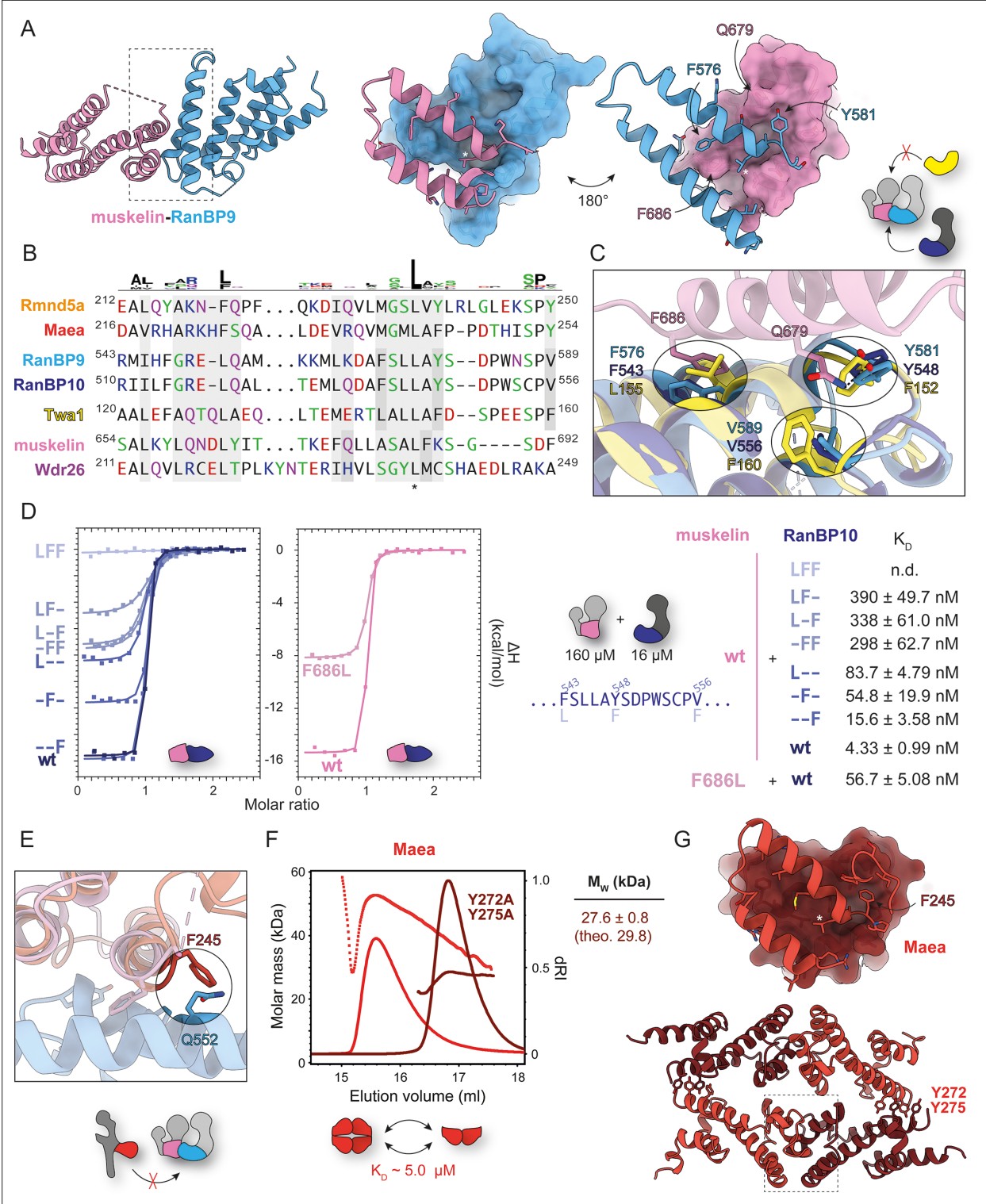

**Figure 2.** Structural determinants of RanBP9/10-muskelin CTLH-CRA specificity. (**A**) 2.0 Å crystal structure of the RanBP9-muskelin CTLH-CRA domains (PDB: 9SNV) with close-up view of the interaction site. (**B**) Sequence alignment of CRA-interface residues highlighting the conserved leucine (marked *). Mouse sequences were used, except for rat muskelin; all are highly similar to the human homologs (*Figure 1—figure supplement 1*). (**C**) Molecular details of the RanBP9-muskelin interface overlaid with RanBP10 (PDB: 9SNF) and Twa1 (PDB: 9SOC). (**D**) Isothermal titration calorimetry (ITC) binding analysis of muskelin with various RanBP10 mutants and the muskelin mutant F686L with RanBP10. (**E**) Overlay of the RanBP9-muskelin interface with Maea (PDB: 9SNH). (**F**) Size exclusion chromatography coupled to multi-angle light scattering (SEC-MALS) analysis of Maea and the double mutant (Y272A, Y275A). (**G**) 1.4 Å crystal structure of the Maea CTLH-CRA domain (PDB: 9SNH) showing details of the homodimerization interface.

*Figure 2 continued on next page*

*Figure 2 continued*

The online version of this article includes the following figure supplement(s) for figure 2:

**Figure supplement 1.** Isothermal titration calorimetry (ITC) binding analysis of muskelin and RanBP10.

**Figure supplement 2.** Concentration-dependent oligomerization of Maea and RanBP10 Q519G.

stacking) is shifted one position earlier in the sequence (*Figure 2B*). ITC analysis with Maea was not feasible due to its oligomeric state in solution. Size exclusion chromatography coupled to multi-angle light scattering (SEC-MALS) measurements showed that the Maea CTLH-CRA domain displayed concentration-dependent apparent masses between a dimer (theoretical 30.2 kDa) and a tetramer (theoretical 60.4 kDa) (*Figure 2F*, *Figure 2—figure supplement 2A*). Fitting these data (measurements at 0.3 ml peak maximum *van gen Hassend et al., 2023*) with a one-site binding model yielded an estimated $K_D$ of ~5 µM for dimer-of-dimer formation (*Figure 2—figure supplement 2A*).

Closer inspection of the Maea crystal structure (*Figure 2G*) confirmed a dimer-of-dimers assembly consistent with the SEC-MALS results. Two canonical CTLH-CRA dimers associate via π-π stacking between Y272 and Y275 contributed by all four Maea subunits. Mutation of these residues to alanine abolished tetramerization and yielded a stable dimer in SEC-MALS (*Figure 2F*). This Maea homodimerization mode had not been described previously, as, in the context of the full CTLH complex, it is disrupted in favor of Twa1 binding.

## Rewiring the binding specificity of RanBP10 from muskelin to Maea

To test whether the steric clash between RanBP10 Q519 and Maea F245 is sufficient to prevent binding, we designed a Q519G mutant with reduced steric hindrance (R10_G). We then combined this substitution with the muskelin-binding-deficient triple mutant (R10_GLFF) to shift RanBP10's binding preference completely from muskelin to Maea.

We analyzed these mutants by SEC-MALS (*Figure 3A*), ITC (*Figure 3B*), and NAGE (*Figure 3C*). Consistent with our hypothesis, R10_G bound both muskelin and Maea, whereas R10_GLFF bound Maea only. SEC-MALS showed that the muskelin CTLH-CRA domain is predominantly monomeric (e.g. $M_W = 18.1 \pm 0.5$ kDa, theoretical 17.9 kDa) with a small dimeric fraction (e.g. $M_W = 34.4 \pm 0.9$ kDa). This weak dimerization occurs independently of the canonical CTLH-CRA interaction site, as heterodimerization with RanBP10 resulted in a small tetrameric population ($M_W = 61.3 \pm 1.7$ kDa). When Maea interacted with R10_G or R10_GLFF, its homodimerization was disrupted and a 1:1 complex was observed ($M_W = 31.8 \pm 1.1$ kDa for R10_G, $M_W = 28.3 \pm 0.8$ kDa for R10_GLFF).

SEC-MALS also revealed that the apparent molecular masses of R10_G and R10_GLFF were elevated compared to wild-type RanBP10 CTLH-CRA (*Figure 3A*, *Figure 2—figure supplement 2B and C*). The introduction of the Q519G mutation appeared to promote weak dimerization. Additional SEC-MALS measurements at different concentrations showed a concentration-dependent dimerization of R10_G (*Figure 2—figure supplement 2B*), with an estimated $K_D$ of ~18 µM.

To understand the structural basis of this behavior, we solved a 1.7 Å crystal structure of the R10_GLFF mutant (*Figure 3D*), which revealed no major changes in the overall fold compared to wild-type RanBP10. As expected, the Q519G substitution relieved steric clashes, facilitating Maea binding and weak RanBP10 dimerization. Comparison of asymmetric units showed that in R10_GLFF, the Y548F substitution reaches deeper into the interface than wild-type Y548, thereby creating closer contacts consistent with weak dimer formation (*Figure 2—figure supplement 2D*).

To validate the proposed binding model of Maea to the RanBP10 mutant with swapped specificity, we solved a 3.5 Å crystal structure of the R10_GLFF-Maea complex (*Figure 3E*). Overlay with the RanBP9-muskelin complex (*Figure 3F*) confirmed the expected effects of the mutations: R10_GLFF no longer forms the π-π-stacking or the hydrogen bond with muskelin and exhibits reduced steric hindrance toward Maea due to the Q519G exchange. Unexpectedly, the interaction with Maea is further stabilized by an additional hydrogen bond between Maea Y254 and R10_GLFF Q540, thus explaining its exclusive binding to Maea.

## Rewiring the binding specificity of Twa1 from Maea to muskelin

We next tested whether the binding specificity of the Twa1 CTLH-CRA domain could be shifted toward a RanBP10-like behavior – binding muskelin instead of Maea. In total, 52 interface mutants

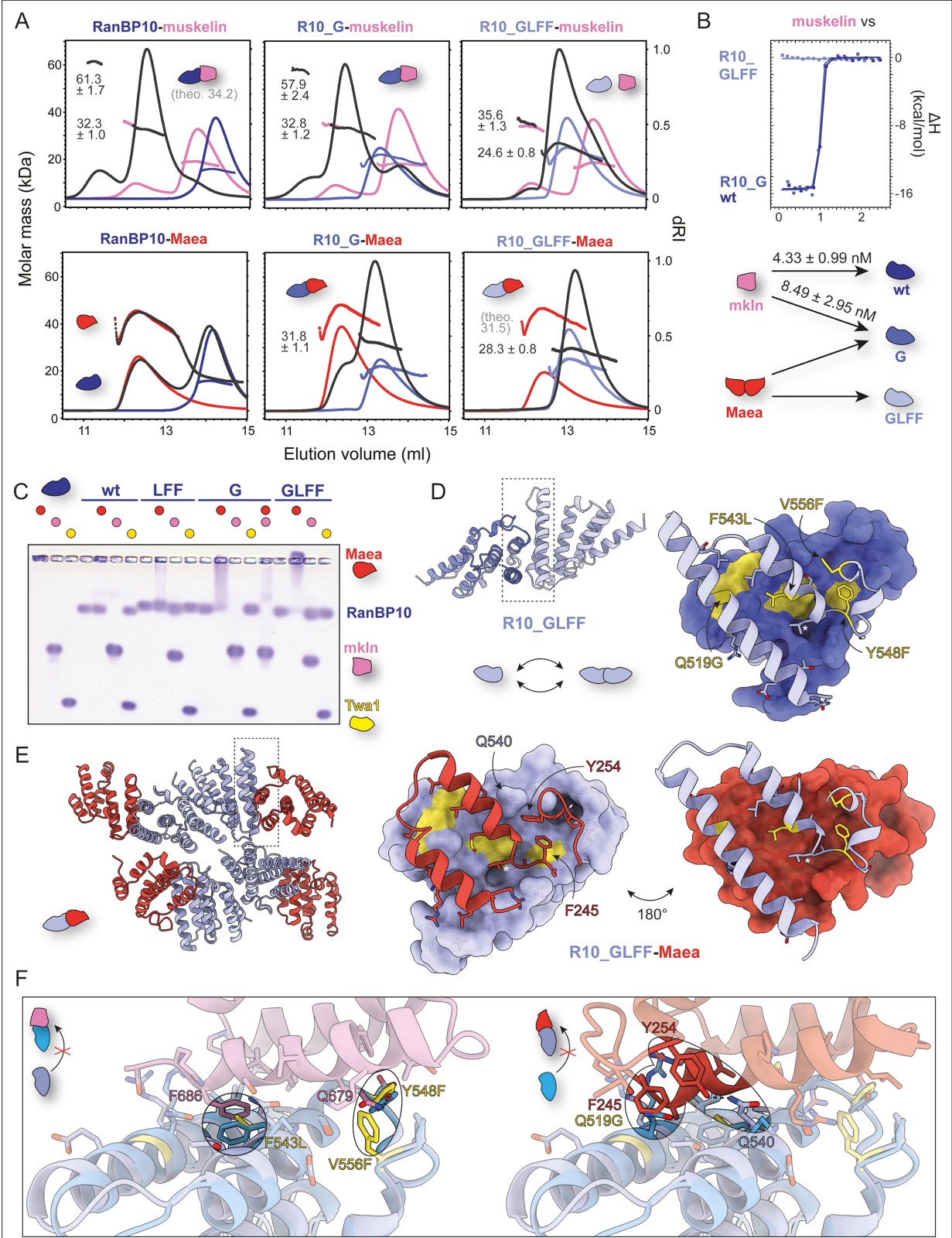

**Figure 3.** Rewiring the binding specificity of RanBP10 from muskelin to Maea. (**A**) Size exclusion chromatography coupled to multi-angle light scattering (SEC-MALS), (**B**) isothermal titration calorimetry (ITC), and (**C**) native agarose gel electrophoresis (NAGE) binding analysis of CTLH-CRA domains of muskelin and Maea with wild-type RanBP10, R10_G (Q519G), and R10_GLFF (Q519G, F543L, Y548F, V556F) mutants. (**D**) 1.7 Å crystal structure of the R10_GLFF CTLH-CRA domain (PDB: 9SNG) showing details of the homodimerization interface. Twa1-like mutations are highlighted in yellow. (**E**) 3.5 Å

*Figure 3 continued on next page*

crystal structure of the R10_GLFF-Maea CTLH-CRA complex (PDB: 9SOH) and overlay (**F**) with the RanBP9-muskelin CTLH-CRA complex (PDB: 9SNV), illustrating the determinants of binding specificity.

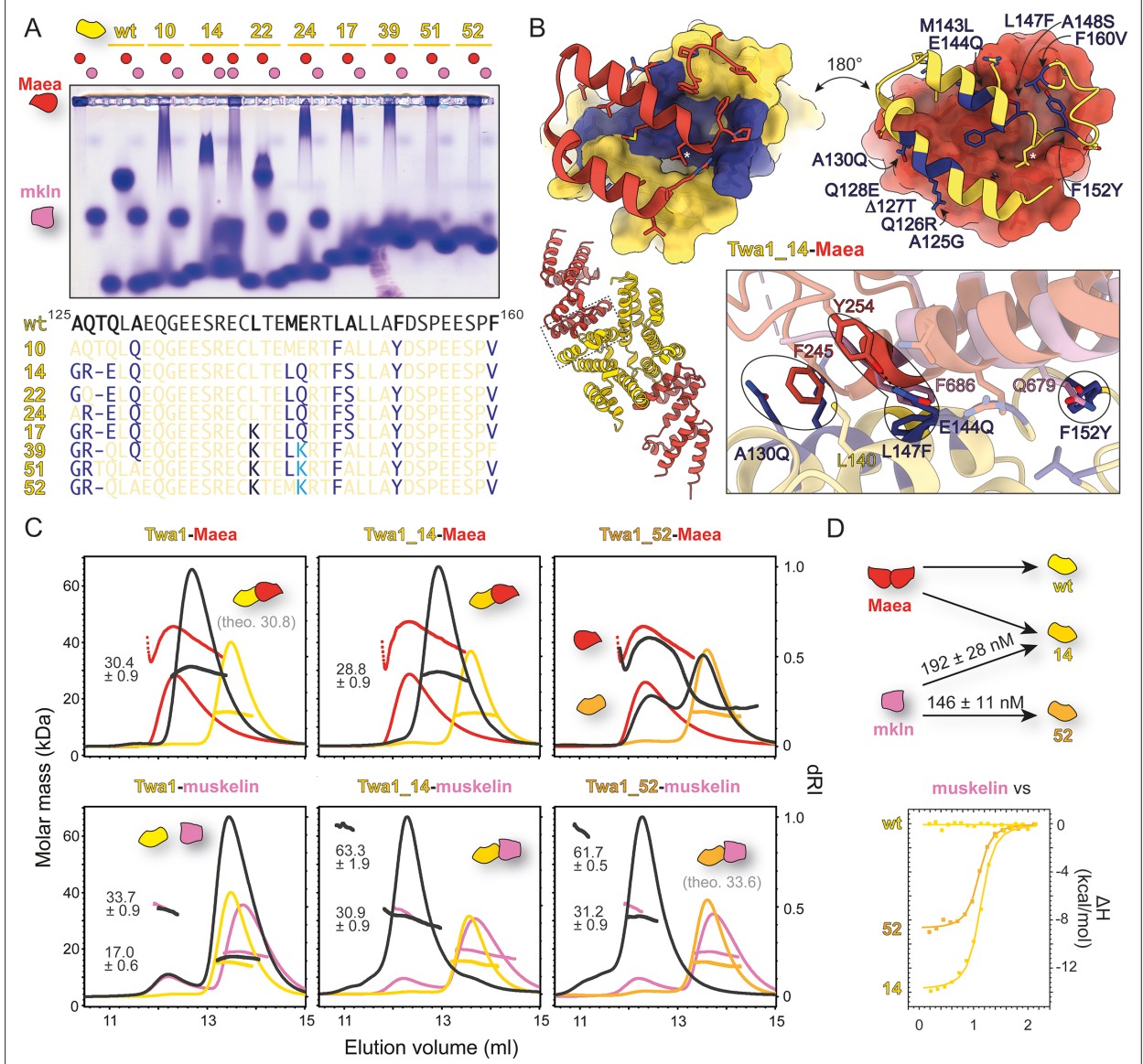

**Figure 4.** Rewiring the binding specificity of Twa1 from Maea to muskelin. (**A**) Native agarose gel electrophoresis (NAGE) binding analysis of various Twa1 mutants to Maea and muskelin, accompanied by a depiction of Twa1 sequence changes. The wild-type sequence corresponds to mouse Twa1 (identical to the human ortholog in the interface region) (**B**) 3.3 Å crystal structure of the Twa1_14-Maea complex (PDB: 9SOI) showing molecular details of the interface, overlaid with muskelin (PDB: 9SNV). (**C**) Size exclusion chromatography coupled to multi-angle light scattering (SEC-MALS) analysis of wild-type Twa1, Twa1_14, and Twa1_52 in complex with Maea or muskelin. (**D**) Isothermal titration calorimetry (ITC) analysis demonstrating that Twa1_14 and Twa1_52 bind muskelin, in contrast to wild-type Twa1.

The online version of this article includes the following figure supplement(s) for figure 4:

**Figure supplement 1.** Sequence overview of Twa1 CTLH-CRA domain mutants.

**Figure supplement 2.** Native agarose gel electrophoresis (NAGE) binding analysis of Twa1 CTLH-CRA domain mutants.

**Figure supplement 3.** Isothermal titration calorimetry (ITC) binding analysis of muskelin and Twa1 mutants.

were generated (*Figure 4—figure supplement 1*) and screened for binding by NAGE (*Figure 4—figure supplement 2*, *Figure 4A*), which provided a rapid readout.

Direct transfer of four key residues from the RanBP10 swap mutant R10_GLFF into Twa1 (Twa1_10: A130Q, L147F, F152Y, and F160V; *Figure 4A*) proved insufficient: muskelin binding was not achieved and Maea interaction was only weakened. This suggested that additional determinants were required. Expanding the RanBP10-like substitutions produced the mutant Twa1_14 (A125G, Q126R, Δ127T, Q128E, A130Q, M143L, E144Q, L147F, A148S, F152Y, and F160V), which retained Maea binding, yet also gained the ability to interact with muskelin.

To understand why Maea binding persisted, we solved a 3.3 Å crystal structure of the Twa1_14-Maea complex (PDB: 9SOI; *Figure 4B*), which revealed that A130Q did not create the expected steric hindrance seen for Q551/Q519 in RanBP9/10 (*Figure 3F*). Since the corresponding α-helix (first CRA helix) in Twa1 is shorter and kinked, A130Q rotates outward, leaving space for Maea F245 (*Figure 4B*). In addition, Maea binding was stabilized by a hydrogen bond between Y254 and the introduced E144Q, analogous to the contact observed in the R10_GLFF-Maea complex (*Figure 3F*).

Stepwise mutagenesis highlighted two substitutions as particularly critical for muskelin interaction: A125G and Q126R (compare Twa_24, Twa1_22; *Figure 4A*; *Figure 4—figure supplement 2*). A125G likely prevents steric interference with L147F, which is essential for correct positioning of the π–π stacking with muskelin, while Q126R probably stabilizes muskelin backbone contacts. Additional substitutions further shifted specificity: L140K (compare Twa1_17; *Figure 4A and B*) introduced steric hindrance against the back-folded Maea loop, whereas E144K (RanBP9-like; compare Twa1_39; *Figure 4A*; *Figure 4—figure supplement 2*) abolished the stabilizing hydrogen bond to Maea Y254. Other substitutions, including Q128E and A148S, were dispensable. F160V was initially removed (Twa1_25; *Figure 4—figure supplement 2*) but later reintroduced for its beneficial effect.

Further changes, removal of T127 (Δ127T), as well as the A130Q and M143L substitutions (Twa1_45) finally eliminated Maea binding but reduced stability and yields. Stability was restored by reintroducing M143L (Twa1_51) or Δ127T (Twa1_52). Both variants bound muskelin, but Twa1_52 showed higher purification yields – comparable to wild-type Twa1 – and was therefore used for further biophysical analyses.

SEC-MALS confirmed a defined 1:1 Twa1_52-muskelin complex ($M_W$ = 31.2 ± 0.9 kDa, theoretical 33.6 kDa). ITC (*Figure 4D*; *Figure 4—figure supplement 3*) revealed an affinity of $K_D$ = 146 ± 11 nM, slightly stronger than Twa1_14 ($K_D$ = 192 ± 28 nM). Interestingly, binding evolved from being strongly enthalpy-driven in early mutants toward a more entropically favored interaction, consistent with reduced conformational strain at the interface.

In summary, wild-type Twa1 binds exclusively Maea, Twa1_14 displayed dual specificity, and Twa1_52 fully switched its specificity to muskelin. Eight mutations – A125G, Q126R, Δ127T, L140K, E144K, L147F, F152Y, and F160V – were required to achieve this specificity swap, twice as many as in RanBP10. This reflects distinct geometric constraints and the presence of fewer residues that could readily engage in alternative interactions.

## Picomolar RanBP9/10 binding to muskelin and Wdr26

Given the similarity between the CTLH-CRA domains of RanBP9 and RanBP10, these subunits are assumed to be interchangeable within the CTLH complex. To test this in the context of near full-length proteins and higher-order assembly, we performed analytical gel filtration binding studies (*Figure 5A*). Interaction of muskelin or Wdr26 with either RanBP9-Twa1 or RanBP10-Twa1 complexes showed no detectable difference, indicating a highly similar assembly behavior.

We next examined whether the binding preference of the engineered Twa1_52 mutant, which was initially tested only against muskelin, also extended to Wdr26. Indeed, tag-free co-expression of Twa1_52 with either the CTLH-CRA domain of muskelin or Wdr26, followed by SEC-MALS analysis, confirmed the formation of well-defined 1:1 complexes in both cases (*Figure 5B*). This demonstrates that the specificity swap in Twa1_52 applies not only to muskelin but also to Wdr26.

Because Twa1_52 binds both muskelin and Wdr26, it provided a practical tool to indirectly quantify the extremely tight interactions of RanBP9 and RanBP10. Although ITC measurements of RanBP9/10 with muskelin or Wdr26 could be performed, the resulting isotherms were too steep for reliable fitting due to sub-nanomolar affinities (*Figure 2—figure supplement 1*). We therefore used Twa1_52 in competition assays: first, Twa1_52 was titrated against muskelin or Wdr26 to form a measurable

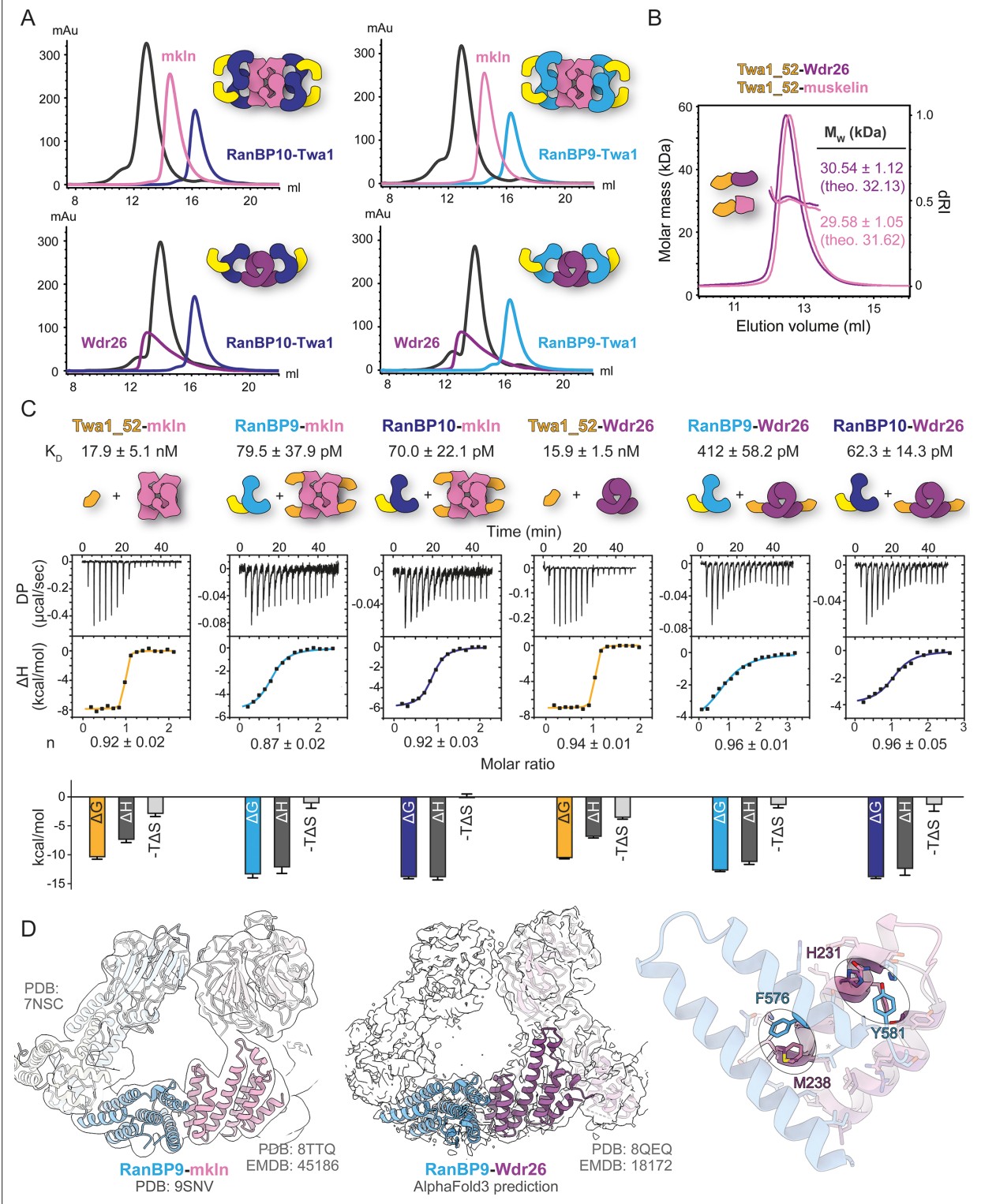

**Figure 5.** Picomolar RanBP9/10 binding to muskelin and Wdr26. (**A**) Analytical size exclusion chromatography of RanBP9/10-Twa1 complexes in the presence of muskelin or Wdr26. (**B**) Size exclusion chromatography coupled to multi-angle light scattering (SEC-MALS) analysis of co-expressed CTLH-CRA domain complexes Twa1_52-Wdr26 and Twa1_52-muskelin. Theoretical $M_W$ is denoted (theo.). (**C**) Isothermal titration calorimetry (ITC) binding analysis of Twa1_52 to muskelin/Wdr26, followed by competitive titration with RanBP9/10-Twa1 complexes. $K_D$ values and stoichiometries (n) are indicated together with the corresponding enthalpic and entropic contributions (below). (**D**) Cryo-EM maps of the mammalian CTLH complex fitted with available structural data highlight the CTLH-CRA dimerization interface in the context of the full-length assembly. An AlphaFold3 prediction of the RanBP9-Wdr26 CTLH-CRA domain overlaid with the RanBP9-muskelin structure (PDB: 9SNV) illustrates their conserved mode of specificity.

complex, and then RanBP9 or RanBP10 was added to displace Twa1_52 (*Figure 5C*). Fitting a competitive binding model using parameters from the initial titration allowed determination of $K_D$ values for RanBP9/10 binding to muskelin and Wdr26.

Twa1_52 CTLH-CRA domain bound tetrameric muskelin and dimeric Wdr26 with similar affinities ($K_D$ = 17.9 ± 5.1 nM and $K_D$ = 15.9 ± 1.5 nM). These $K_D$ values were approximately tenfold lower than those measured in the reverse titration of the isolated muskelin CTLH-CRA domain against Twa1_52 (*Figure 4D*). Both RanBP9 and RanBP10 outcompeted Twa1_52 binding and bound muskelin very tightly, with comparable affinities ($K_D$ = 79.5 ± 37.9 pM and 70.0±22.1 pM). RanBP10 bound also Wdr26 with similar affinity ($K_D$ = 62.3 ± 14.3 pM), whereas RanBP9 bound Wdr26 slightly weaker ($K_D$ = 412 ± 58.2 pM) than muskelin.

SEC-MALS analysis of ITC-derived RanBP9-Twa1 complexes containing muskelin or Wdr26 confirmed a 1:1 binding stoichiometry of RanBP9 with either partner (*Figure 1—figure supplement 3D*), consistent with previous assembly studies (*van gen Hassend et al., 2023*). Existing cryo-EM data of the mammalian CTLH complex fit well with our RanBP9-muskelin CTLH-CRA structure (*Figure 5D*) and with the AlphaFold3 prediction (*Abramson et al., 2024*) of the RanBP9-Wdr26 interface (crystals did not diffract). These results suggest that additional contacts between the SPRY domain of RanBP9/10 and the kelch domain of muskelin or the WD40 domain of Wdr26 contribute to positioning and tighter interaction. This may explain also the much tighter affinity in the mid-picomolar range compared with the ~100-fold weaker nanomolar interaction observed in the reverse titration of the isolated CTLH-CRA domains.

Overlay of the AlphaFold3 prediction of Wdr26 CTLH-CRA domain with the RanBP9-muskelin structure showed that Wdr26 engages the same region of RanBP9 and uses similar contact principles. The predicted interface includes a Met-π interaction between Wdr26 M238 and RanBP9 F576 and a hydrogen bond between Wdr26 H231 and RanBP9 Y581, consistent with the comparable affinities of both complexes.

## Discussion

How do you build a megadalton protein ring? The CTLH complex achieves this with helical scaffold proteins that have two distinct dimerization interfaces. Both interfaces – the LisH domain (*Delto et al., 2015*; *Emes and Ponting, 2001*) and the CRA dimerization interface – consist of two α-helices dominated by hydrophobic residues at their core, such as leucine and phenylalanine in the LisH domain (*Figure 1—figure supplement 2*) or a conserved leucine in the CRA interface (*Figure 6*, marked with *). Specificity arises from additional interactions such as specific hydrogen bonds or π-π stacking. The geometries of these interfaces, however, differ: CRA helices align nearly parallel, so that adjacent CRA dimers contact each other almost orthogonally, whereas LisH dimers adopt an open angle of about 30°, with the second helix running almost parallel while the first crosses its partner. Additional structural elements stabilize both helix pairs: LisH dimerization is reinforced by a leucine-rich cross-over helix (CRA$^C$), whereas CRA dimerization is supported by a loop segment following the second helix that folds back and contributes further contacts (*Chrustowicz et al., 2024*; *Qiao et al., 2020*; *Sherpa et al., 2021*).

While many ring-shaped assemblies – such as sliding clamps, ion channels, pore complexes, chaperonins, AAA+ ATPases, and even viral capsids – are formed by homooligomerization, the CTLH complex uses heterooligomerization. Although a CTLH-like geometry could, in principle, be built from a single oligomerizing subunit, such an assembly would be unsuitable for a RING E3 ligase, which must simultaneously provide adaptor interfaces for E2~Ub and recognition surfaces for substrates positioned at defined sites. Achieving this spatial choreography requires heterotypic subunits that use the same assembly principle but deploy distinct domains around the ring – allowing form to follow function.

Our crystal structures of CTLH-CRA domains provide high-resolution insights into otherwise, at least when expressed in *E. coli*, insoluble proteins such as RanBP9/10 and Maea. Maea homodimerization, reported here for the first time, is compatible with the full-length protein but would prevent ring formation and therefore likely represents an intermediate state in the assembly. The previously uncharacterized RanBP9-muskelin interface revealed how two CTLH-CRA domains interact. Using these structural data, we engineered CTLH-CRA domains with swapped binding specificity:

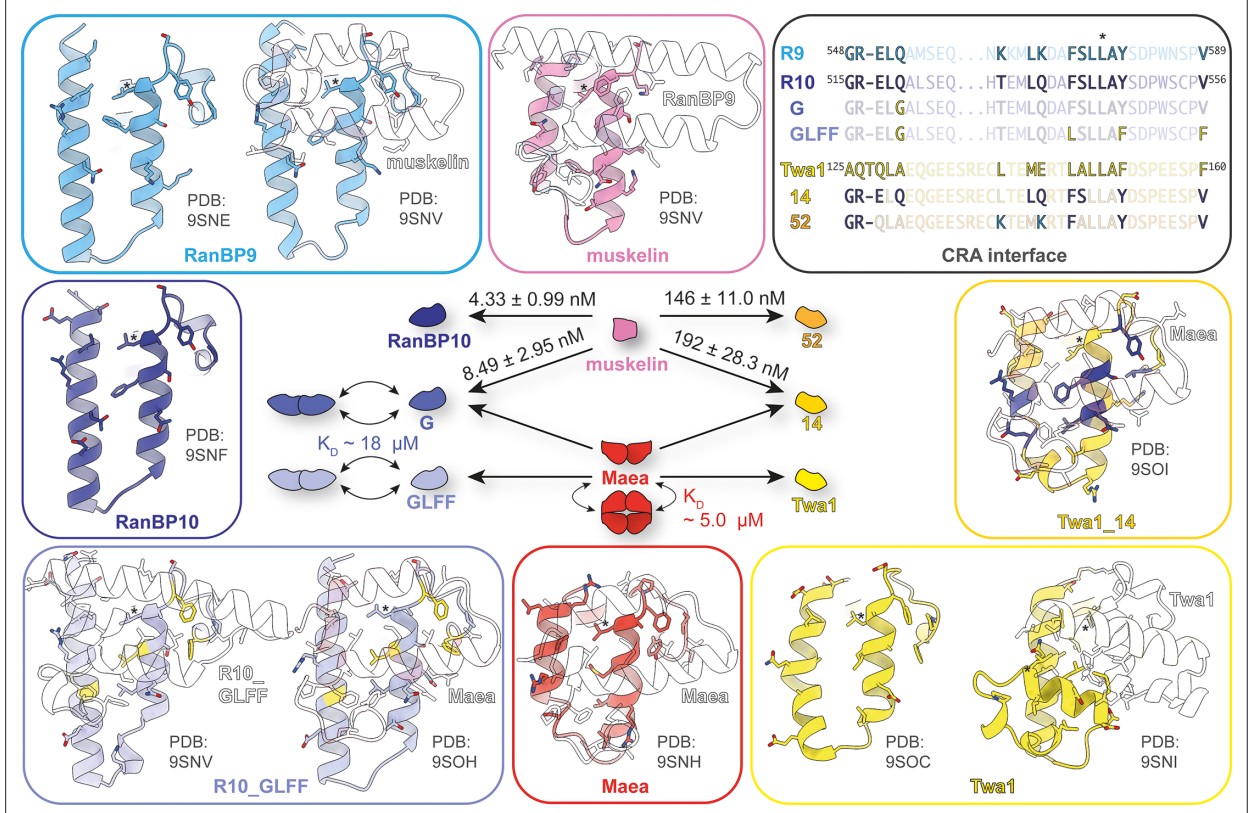

**Figure 6.** CTLH-CRA domain specificity code. Crystal structures determined in this study of CTLH-CRA domains highlight the specificity of the interaction interfaces and the position of the conserved leucine residue (*) for comparison. Mutational analyses demonstrate that binding specificity can be rewired bidirectionally: the RanBP10 quadruple mutant R10_GLFF (Q519G, F543L, Y548F, and V556F) successfully switched its preference from muskelin to Maea, while the engineered Twa1_52 mutant (A125G, Q126R, Δ127T, L140K, E144K, L147F, F152Y, and F160V) acquired RanBP10-like specificity toward muskelin. Together, these structures and functional swaps define the residue-level code underlying partner recognition within CTLH-CRA interfaces.

The online version of this article includes the following figure supplement(s) for figure 6:

**Figure supplement 1.** Overlay of AlphaFold3-predicted CTLH-CRA interfaces across species.

**Figure supplement 2.** CTLH-CRA interface of the yeast Gid1-Gid7.

**Figure supplement 3.** Non-CTLH complex CTLH-CRA domains.

**Figure supplement 4.** Electrostatic surface representation of CTLH-CRA domain structures determined in this study.

A RanBP10 mutant that acquired a Twa1-like preference for Maea instead of muskelin, and Twa1 mutants that adopted a RanBP9/10-like preference for muskelin instead of Maea.

Importantly, swapping the specificity of these very tight interfaces (nM-pM range) is substantially more challenging than merely disrupting them. Charged substitutions introduced into the hydrophobic core readily abolished dimerization – e.g., the Twa1_3 mutant (L122R, A151R) no longer bound Maea (*Figure 4—figure supplement 2*), or the pathogenic human Wdr26 S254R variant (S234R in mouse), which causes, by impaired complex formation (*Gross et al., 2024*), Skraban-Deardorff syndrome (*Skraban et al., 2017*; *Tickle et al., 2016*), an intellectual disability disorder. By contrast, successful switching requires a coordinated loss of the original contacts together with the formation of new partner-specific interactions, while preserving the rather unspecific hydrophobic core architecture.

For RanBP10, relieving a single clash by substituting Q519 with glycine (R10_G) was sufficient to permit Maea binding, and in combination with F543L, Y548F, and V556F, to abolish muskelin binding (*Figure 6*). Both R10_G and R10_GLFF bound Maea but not Twa1, even though the Q519G substitution should also alleviate a clash with Twa1, which – like Maea – carries a phenylalanine at the equivalent position. The difference may be explained by the newly formed hydrogen bond in the R10_GLFF-Maea complex (*Figure 3F*), in which Maea Y254 interacts with RanBP10 Q540, whereas

Twa1 F160 cannot. Together with additional subtle steric constraints, this likely accounts for the Maea specificity of R10_GLFF.

In contrast, reversing the specificity in Twa1 required more extensive remodeling – eight rather than four substitutions (*Figure 6*) – reflecting differences in overall geometry, such as the kink in the first CRA helix and the absence of pre-existing residues that could engage directly in the new interactions, like RanBP10 Q540. Engineered Twa1_52 mutant bound (near) full-length muskelin and Wdr26 comparably but was outcompeted by its native partners RanBP9 and RanBP10. Interestingly, RanBP9 appears to bind Wdr26 approximately tenfold weaker than RanBP10, possibly explaining why RanBP10 can substitute for RanBP9 in Wdr26-containing CTLH complexes during erythrocyte maturation (*Sherpa et al., 2022*), although the functional significance of this remains unclear. Despite this difference, both proteins likely assemble similarly owing to their nearly identical interfaces.

AlphaFold3 predictions (*Abramson et al., 2024*) of CTLH-CRA dimerization interfaces across model organisms indicate that the determinants of binding specificity are broadly conserved beyond mammals (*Figure 6—figure supplement 1*). In muskelin-containing species such as *X. laevis* and *D. rerio*, the RanBP9/10 interface is particularly well preserved, employing the same mechanism of specificity as in mouse: π–π and Met–π interactions involving RanBP9/10 phenylalanine residues, and a hydrogen bond formed by RanBP9/10 tyrosine with muskelin glutamine or Wdr26 histidine. Remarkably, this hydrogen bond is conserved even between the *A. thaliana* homologs of RanBP9 and Wdr26.

By contrast, *C. elegans* Wdr26 contains an insertion at the interface predicted to form β-strands, resulting in an AlphaFold3 prediction with low interface confidence. In yeast, the Gid1 (RanBP9 homologue) interface is largely conserved, except for the missing phenylalanine (*Figure 6—figure supplement 2*; PDB: 7NSB). The Gid1 N-terminus, however, forms an additional contact site with the Wdr26 homolog Gid7, a secondary interaction that likely relaxed or shifted the constraints governing interface specificity.

Core-specific determinants, such as the conserved leucine residue (*), are also present in other CTLH-CRA domains such as SMU1, Wdr47, TOPLESS, and TRP2 (*Figure 6—figure supplement 3A*). However, the position of a conserved aromatic residue distinguishes different interface classes (*Figure 6—figure supplement 3B*). Muskelin uses its phenylalanine, immediately following the conserved leucine*, for π-π stacking with RanBP9, whereas other CTLH complex members carry their conserved Phe/Tyr two residues after the leucine*. This aromatic residue is crucial for specific interactions – e.g., Maea F245 not only discriminates against RanBP9/10 binding but also stabilizes the loop that folds back to form a hydrogen bond with Y254. By contrast, non-CTLH complex CTLH-CRA domains have their aromatic residue shifted seven positions upstream to leucine*, resulting in a loop that does not fold back into the interface. A comparison of the Maea and TRP2 homodimerization interfaces (*Figure 6—figure supplement 3B*) showed that this positional shift relocates the aromatic residue to the opposite side of the interface, reflecting positional rather than sequence conservation.

Within the CTLH complex subunits, LisH-CRA[C] and CTLH-CRA domains exclusively mediate ring assembly – except for Twa1. Twa1 can additionally interact via its CTLH-CRA domain with Armc8 through electrostatic complementarity, i.e., the negatively charged surface of Twa1 (*Figure 6—figure supplement 4*) engages positively charged patches on Armc8 (*Sherpa et al., 2021*). Outside the CTLH complex, CTLH-CRA domains have hydrophobic grooves adjacent to the dimerization interface that mediate protein interactions (*Figure 6—figure supplement 3C*). For example, SMU1 binds RED on the N-terminal side of the interface (*Ashraf et al., 2019*), whereas TRP2 binds IAA1, NINJA, and EAR-motif peptides on the C-terminal side (*Ke et al., 2015*). These observations suggest that such binding grooves are either absent in CTLH complex subunits or may represent an underexplored protein interaction site (*Figure 6—figure supplement 4*), potentially explaining structural differences of the RanBP9 and RanBP10 paralogs which behave similarly in the assembly.

The CTLH complex exemplifies how evolution can elaborate a simple helical scaffold into a versatile assembly platform. Subtle variations in aromatic and polar residues fine-tune partner specificity, giving rise to distinct LisH-CRA[C] and CTLH-CRA interfaces. By mapping this residue-level code and demonstrating bidirectional specificity rewiring of CTLH-CRA domains, we provide a framework for understanding and rationally manipulating CTLH complex ring assembly.

Mutations such as R10_LFF or Twa1_3, which selectively disrupt CRA-CRA dimerization, provide powerful tools to dissect assembly-dependent functions. Unlike protein depletion or knockout strategies, targeted interface mutations selectively perturb defined subunit interactions while preserving

other structural or regulatory roles. While R10_LFF blocks the specificity determinants identified in this manuscript, the conserved nature of the CRA interface, exemplified by Twa1_3 (L122R, A151R), suggests a broader strategy. Substitution of analogous hydrophobic residues with charged residues (see alignment, *Figure 2B*) in other core CTLH subunits would be expected to disrupt CRA-mediated interactions in a similar manner.

Such mutations allow selective enforcement of defined assembly routes by disabling alternative ring-forming interactions. For example, mutation of key residues within the muskelin CRA interface (e.g. specificity residues F686 and Q679; *Figure 2*) would prevent muskelin incorporation into the ring, thereby favoring substitution by Wdr26 during assembly. Conversely, mutation of the Wdr26 CRA interface would promote incorporation of muskelin. Combining these CRA-interface mutations with interaction-blocking variants such as R10_LFF would further restrict alternative connectivity and enable generation of ring complexes with defined composition, such as RanBP9-Wdr26 assemblies.

Specificity-switch mutations extend this approach by enabling non-canonical CTLH ring architectures. Introduction of the Twa1_52 mutant may favor incorporation of Wdr26/muskelin dimers at the expense of Rmnd5a-Maea, thus resulting in assemblies lacking catalytic modules. Conversely, the R10_GLFF mutation may promote incorporation of additional catalytic modules by replacing Wdr26/muskelin dimers. Such complexes would lack Wdr26/muskelin-mediated substrate recruitment and may therefore rely more strongly on the Gid4-Armc8 recognition route. When both switching mutations are combined, the orientation of the RanBP9/10-Twa1 module within the ring may be inverted, potentially altering the spatial positioning of domains such as the RanBP9/10 SPRY domain and thereby influencing substrate engagement via the Gid4-Armc8 axis.

Collectively, these findings define a residue-level structural code governing CTLH ring assembly and demonstrate that complex architecture can be rationally manipulated. The ability to selectively eliminate, redirect, or reorient subunit interactions may help to uncover substrate-specific dependencies and could serve as blueprints for designing other ring-shaped complexes beyond E3 ligases.

## Methods
### Cloning
Domain boundaries of the CTLH-CRA domain constructs were defined based on AlphaFold2 predictions (*Varadi et al., 2022*) of murine RanBP9 (P69566), RanBP10 (Q6VN19), Twa1 (Q9D7M1), Wdr26 (Q8C6G8), Maea (Q4VC33), and Rmnd5a (Q80YQ8), as well as rat muskelin (Q99PV3). All constructs were cloned using the SLIC method. RanBP9 (including the CRA$^C$ segment; residues 324–638, Δ391–534) and RanBP10 (315–608, Δ382–529) were cloned behind an N-terminal 6xHis-SUMO tag in a pETM-SUMO vector and, for tagless co-expression, in a pCDF-Duet1 vector. RanBP10 (315–608, Δ382–529), Twa1 (58-190), Wdr26 (136-278), Maea (156-279), Rmnd5a (147-275), and muskelin (C-terminally shortened: 205–719, Δ248–624) were cloned into a pETM-14 vector (kindly provided by Dr. Florian Sauer) containing an N-terminal 6xHis-tag, followed by a 3C protease cleavage site. Mutant constructs included RanBP10 (Q547G, F571L, Y576F, V584F, and combinations thereof), muskelin (F686L, 205–735 Δ248–625), and Twa1 (Twa1_1–52; see *Figure 2—figure supplement 2*), were all similarly cloned into the pETM-14 vector. Twa1_52 and RanBP9 (324–614, Δ391–534) were additionally cloned into the pCDF-Duet1 vector for tagless co-expression.

### Recombinant protein expression and purification
Proteins were expressed in *E. coli* BL21 CodonPlus (DE3) RIL cells grown in LB medium supplemented with 34 μg/ml chloramphenicol and 50 μg/ml kanamycin. For co-expression, 50 μg/ml streptomycin was additionally included. Cultures were grown at 37°C to an optical density (OD$_{600}$) of 0.6–1.2, and protein expression was induced with 0.5 mM isopropyl β-D-1-thiogalactopyranoside. Expression continued overnight at 20°C.

The RanBP9 (69–653)-Twa1 (1–228) complex, Twa1 (1–228), Wdr26 (102–641), muskelin (1–735), and muskelin's CTLH-CRA domain (205–735 Δ248–625) were expressed and purified as described previously (*van gen Hassend et al., 2023*). The RanBP10 (57–648)-Twa1 (1–228) complex, as well as SUMO-tagged RanBP9 and RanBP10 CTLH-CRA domains, were purified following the same procedure established for the RanBP9-Twa1 complex.

All remaining constructs and co-expressed CTLH-CRA complexes were purified using a two-step protocol: initial Ni-affinity capture followed by SEC. For lysis, cells were resuspended in buffer (20 mM tricine pH 8.0, 500 mM NaCl, 1 mM tris(2-carboxyethyl)phosphine (TCEP)) supplemented with DNaseI (AppliChem), EDTA-free cOmplete protease inhibitor cocktail (Roche), and lysozyme (Carl Roth). Cells were disrupted by two passages through a Microfluidizer (LM20, Microfluidics) at 1.5 kbar or by sonification (SONOPLUS HD 3100 homogenizer with VS 70T probe, Bandelin) at 82% amplitude, 1 s on/off pulses, for at least 2 min per 50 ml lysate. Lysates were clarified by centrifugation at 38,000 × $g$ for at least 30 min and loaded onto Protino Ni-IDA resin equilibrated with buffer. Beads were washed three times with 10 column volumes of buffer. The 6xHis-tag was cleaved overnight with 3C protease under gentle shaking. Proteins were eluted in buffer, concentrated, and subjected to SEC using either a Superdex 75 increase 10/300 GL column (Cytiva) or a HiLoad 16/600 Superdex 75 pg (Cytiva) on an ÄKTA pure purification system (Cytiva). Suitable elution fractions were combined and either used directly for further analysis or stored at –80°C. For later usage, proteins were thawed, centrifuged at 25,000 × $g$ for 30 min, and the concentration was determined using UV absorption at 280 nm and calculated molar extinction coefficients.

### Analytical SEC

RanBP9-Twa1, RanBP10-Twa1, muskelin, and Wdr26 were diluted to a final concentration of 30 µM with a sample volume of 500 µl and used either as control samples or for complex assembly at a 1:1 molar ratio. Samples were incubated on ice for 30 min, centrifuged at 30,000 × $g$ for 20 min, and applied to a Superose 6 increase 10/300 GL column (Cytiva) pre-equilibrated with buffer.

### Native agarose gel electrophoresis

CTLH-CRA domains were diluted in buffer to a final concentration of 200 µM and incubated with their prospective binding partner for at least 30 min at room temperature. 4–10 µl samples were separated on freshly prepared 0.8% HEEO agarose gels for 2–3 hr at 120 V and 4°C in NAGE buffer (25 mM Tris and 200 mM glycine). Following electrophoresis, gels were incubated in staining solution (0.05% Coomassie R250, 50% methanol, and 10% acetic acid) for at least 20 min, followed by destaining (10% ethanol, 5% acetic acid).

### Size exclusion chromatography coupled to multi-angle light scattering

Suitable SEC columns were equilibrated with freshly prepared buffer, and 100 µl of protein sample was separated using an ÄKTA go chromatography system (Cytiva). Differential refractive index and laser scattering signals were recorded with a DAWN HELEOS 8+ light scattering detector and an Optilab T-rEX refractive index detector (both from Wyatt Technology) coupled in-line downstream to the chromatography system. Molar masses were calculated and plotted using the Astra 6.1.5 software (Wyatt Technology).

CTLH-CRA domains and complexes, like RanBP10 and Twa1 mutants with Maea or muskelin, were prepared at a final concentration of 250 µM and separated on a Superdex 75 10/300 GL column (Cytiva). Larger proteins and complexes, like Twa1, co-expressed Twa1-Rmnd5a (CTLH-CRA), Twa1-Maea (CTLH-CRA), as well as the Maea CTLH-CRA dilution series and Maea double mutant (Y272A, Y275A), were analyzed on a Superdex 200 10/300 GL column (Cytiva). Even larger assemblies, like the ITC-assembled RanBP9-Twa1-muskelin/Wdr26 complexes, were analyzed on a Superose 6 10/300 GL (Cytiva).

Self-association of Maea CTLH-CRA domain and the RanBP10 Q547G mutant was assessed using dilution series of the respective proteins. Molar masses from 0.3 ml peak fractions were plotted against the concentration of each fraction, and a one-site binding curve was fitted with GraphPad Prism 9.

### Isothermal titration calorimetry

Proteins were dialyzed overnight into freshly prepared buffer (20 mM tricine pH 8.0, 500 mM NaCl, and 1 mM TCEP). After centrifugation, proteins were diluted to suitable concentrations (e.g. 45 µM ↔ 4 µM for RanBP10-Twa1 ↔ muskelin; 160 µM ↔ 16 µM for muskelin ↔ RanBP10 or Twa1 mutants, or RanBP9/10-Twa1 ↔ Twa1_52-muskelin/Wdr26; or 192 µM ↔ 19 µM for Twa1_52 ↔ muskelin/Wdr26). The more highly concentrated protein was loaded into the syringe and the lower-concentrated protein into the cell of an ITC-200 instrument (Malvern Analytics) equilibrated at 25°C. Titrations consisted

of 16 injections, the first of 1.25 µl and the remaining 15 of 2.5 µl each, into their putative binding partners. The differential power required to heat or cool the sample cell to match the reference cell temperature was recorded and analyzed using Microcal software (Malvern Analytics) in Origin, applying a one-site binding model. Titrations were typically repeated three to four times, and standard errors were calculated from these measurements.

For competition experiments, two consecutive titrations were performed in three steps. First, Twa1_52 was titrated to muskelin or Wdr26, and binding parameters were determined using a one-site binding model. Second, after the first experiment, a few microliters were removed from the cell to prevent overfilling caused by the added titrant volume. Third, the competitive binding partner (RanBP9/10-Twa1 complex) was titrated into the pre-formed Twa1_52-muskelin/Wdr26 complex, and the resulting thermograms were analyzed with a competitive binding model using parameters obtained from the first experiment.

## Crystallization, data collection, and structure solution

Crystals of the RanBP10-Twa1 complex were grown at 4°C using a hanging-drop vapor diffusion setup by mixing 2 µl of protein (12 mg/ml) with 2 µl of reservoir solution (100 mM bis-tris 6.0, 20 mM $(NH_3)_6CoCl_3$, 180 mM $MgCl_2$, 9% PEG 3350). For cryoprotection, crystals were transferred into cryo-solution (reservoir composition plus 30% glycerol) and flash-frozen in liquid nitrogen. Diffraction data were collected at the European Synchrotron Radiation Facility (ESRF, Grenoble, France), beamline ID23-2, at a wavelength of 0.8731 Å. Crystals diffracted highly anisotropically.

The RanBP9 (324–638, Δ391–534) CTLH-CRA domain crystals were obtained by sitting-drop vapor diffusion, mixing 300 nl protein (2 mg/ml) with 300 nl reservoir solution (100 mM sodium citrate pH 5.5, 10 mM $SrCl_2$, 20% PEG 3350). Crystals were cryoprotected (reservoir plus 30% glycerol), and data were collected at the Deutsche Elektronen-Synchrotron (DESY, Hamburg, Germany), beamline P14.

The RanBP10 (315–608, Δ382–529) CTLH-CRA domain crystallized using hanging-drop vapor diffusion with a 1:1 mixture (2 µl+2 µl) of protein and reservoir solution (100 mM tris pH 7.8, 15% PEG 3350). Crystals were flash-frozen in cryo-solution (reservoir plus 30% glycerol), and their diffraction was measured at beamline ID30-A3, ESRF.

The Twa1 (58–190) CTLH-CRA domain crystals were grown by mixing 2 µl protein (50 mg/ml) with 2 µl reservoir (100 mM tris 8.0, 1.6 M potassium sodium tartrate) in a hanging-drop vapor diffusion experiment. Crystals were similarly cryoprotected (reservoir plus 30% glycerol) and diffraction data were measured at P14, DESY.

The Maea (156–279) CTLH-CRA domain was crystallized using hanging-drop vapor diffusion of 2 µl protein (2 mg/ml) mixed with 0.5 µl reservoir solution (100 mM sodium acetate pH 4.6, 0.2 M sodium acetate, 23% PEG 4000). Cryoprotected (100 mM sodium acetate pH 4.6, 0.2 M sodium acetate, 30% PEG 400) crystals were measured at P13, DESY.

RanBP9 (324–614, Δ391–534)-muskelin (205–719, Δ248–624) CTLH-CRA domain complex crystals were obtained in a sitting-drop vapor diffusion experiment where 300 nl protein (24 mg/ml) was mixed with 150 nl reservoir (100 mM tris 8.5, 10 mM $NiCl_2$, 20% PEG MME 2000) and cryoprotected (100 mM tris 8.5, 10 mM $NiCl_2$, 30% PEG 400). Diffraction data were measured at P14, DESY.

The RanBP10 (Q547G, F571L, Y576F, V584F, 315–608 Δ382–529) CTLH-CRA domain mutant crystals were grown in a sitting-drop vapor diffusion setup using 300 nl protein at 12 mg/ml and 300 nl reservoir (100 mM bis-tris pH 6.5, 20% PEG MME 5000). Cryoprotected (100 mM bis-tris pH 6.5, 30% PEG 400) crystals were measured at P14, DESY.

The RanBP10 (Q547G, F571L, Y576F, V584F, 315–608 Δ382–529)-Maea (156-279) CTLH-CRA domain complex was crystallized in a hanging-drop vapor diffusion experiment by mixing 2 µl protein (22 mg/ml) with 0.5 µl reservoir (100 mM tris pH 8.0, 200 mM NaCl, 20% PEG 6000). Crystals were cryoprotected (100 mM tris pH 8.0, 30% PEG 400), and diffraction data were collected at P14, DESY.

The Twa1_14 mutant (A125G, Q126R, ΔT127, Q128E, A130Q, M143L, E144Q, L147F, A148S, F152Y, F160V; 58–190)-Maea (156-279) CTLH-CRA complex crystals were obtained by mixing 2 µl protein (15 mg/ml) with 2 µl reservoir (100 mM hepes 7.0, 1 M sodium malonate, 0.5% Jeffamine ED2001). Crystals were cryoprotected (reservoir plus 30% glycerol), and diffraction data were measured at P14, EMBL.

Data collection for all crystals was performed at 100 K using the remote setup at the respective synchrotron beamlines. Diffraction data were integrated with XDS (*Kabsch, 2010*), anisotropy-corrected,

and merged using the STARANISO server (*Tickle et al., 2016*). Structures were solved by molecular replacement using AlphaFold2 (*Jumper et al., 2021*; *Varadi et al., 2024*; *Varadi et al., 2022*) or AlphaFold3 (*Abramson et al., 2024*) models of the respective constructs with PHASER of the CCP4 suite interface (*Potterton et al., 2003*). Models were refined with Phenix (*Adams et al., 2010*) and manually adjusted in Coot (*Emsley et al., 2010*). Details of the data collection and refinement can be found in *Table 1*.

### Generation of sequence logos

Sequence logos based on sequence alignments were generated using the online WebLogo service (https://weblogo.berkeley.edu/logo.cgi).

### Generation of AlphaFold3 predicted CTLH-CRA interfaces

Sequences were derived from UniProt entries (denoted in *Figure 4—figure supplement 3*) of the different homologs of the human CTLH core proteins (Rmnd5a, Rmnd5b, Maea, Twa1, RanBP9, RanBP10, muskelin, and Wdr26) and of the model organism *A. thaliana*, *S. cerevisiae*, *D. melanogaster*, *C. elegans*, *D. rerio*, *X. laevis*, *M. musculus*. For interface prediction, only the CTLH-CRA domain was used since full-length protein predictions led to incorrect LisH-CRA[C] interactions. Since RanBP9 and RanBP10 have a flexible linker within the CTLH-CRA domain, this linker was kept for prediction. Likewise, for muskelin predictions, the kelch domain, which inserts into the CTLH-CRA domain, was also kept for prediction, to preserve the CTLH-CRA fold. Domain borders were based on the AlphaFold predictions (*Varadi et al., 2024*) of the respective entries. Sequences were collected and fed to the AlphaFold Server for interface prediction.

## Acknowledgements

This work was supported by the Deutsche Forschungsgemeinschaft (GRK 2243) and the Rudolf Virchow Center to HS.

## Additional information

### Funding

| Funder | Grant reference number | Author |
| --- | --- | --- |
| Deutsche Forschungsgemeinschaft | GRK 2243 | Pia Maria van gen Hassend Hermann Schindelin |
| Rudolf Virchow Center | | Hermann Schindelin |

The funders had no role in study design, data collection and interpretation, or the decision to submit the work for publication.

### Author contributions

Pia Maria van gen Hassend, Conceptualization, Data curation, Formal analysis, Investigation, Visualization, Methodology, Writing – original draft, Writing – review and editing; Hermann Schindelin, Conceptualization, Supervision, Funding acquisition, Investigation, Writing – review and editing

### Author ORCIDs

Pia Maria van gen Hassend ⓘ https://orcid.org/0000-0002-1245-4339
Hermann Schindelin ⓘ https://orcid.org/0000-0002-2067-3187

Reviewer #1 (Public review): https://doi.org/10.7554/eLife.110152.3.sa1
Reviewer #2 (Public review): https://doi.org/10.7554/eLife.110152.3.sa2
Reviewer #3 (Public review): https://doi.org/10.7554/eLife.110152.3.sa3
Author response https://doi.org/10.7554/eLife.110152.3.sa4

**Table 1.** Data collection and refinement statistics of crystal structures from CTLH complex core subunits.

| | RanBP9 CTLH-CRA | RanBP10 CTLH-CRA | R10_GLFF CTLH-CRA | Maea CTLH-CRA | Twa1 CTLH-CRA | Mkln-R9 CTLH-CRA | R10_GLFF-Maea CTLH-CRA | Twa_14-Maea CTLH-CRA | RanBP10-Twa1 |
|---|---|---|---|---|---|---|---|---|---|
| PDB ID | 9SNE | 9SNF | 9SNG | 9SNH | 9SNI | 9SNV | 9SOH | 9SOI | 9SOC |
| Wavelength (Å) | 0.9779 | 0.9677 | 0.9763 | 0.9763 | 0.9763 | 0.9763 | 0.9763 | 0.9763 | 0.8731 |
| Space group | P 1 21 1 | P 2 21 21 | P 21 21 2 | P 21 21 21 | P 41 21 2 | P 61 2 2 | F 41 3 2 | P 32 2 1 | P 61 2 2 |
| a, b, c (Å) | 41.277 80.320 45.017 | 55.331 75.500 76.386 | 52.079 68.629 76.453 | 62.075 76.181 108.149 | 118.843 118.843 56.490 | 84.086 84.086 228.282 | 359.825 359.825 359.825 | 142.346 142.346 213.234 | 124.920 124.920 479.645 |
| α, β, γ (°) | 90.00 101.61 90.00 | 90.00 90.00 90.00 | 90.00 90.00 90.00 | 90.00 90.00 90.00 | 90.00 90.00 90.00 | 90.00 90.00 120.00 | 90.00 90.00 90.00 | 90.00 90.00 120.00 | 90.00 90.00 120.00 |
| Resolution limits (Å) | 44.096–1.810 (2.006–1.810) | 44.810–2.069 (2.294–2.069) | 43.042–1.737 (1.870–1.737) | 48.122–1.431 (1.583–1.431) | 46.882–2.113 (2.282–2.113) | 44.919–1.965 (2.150–1.965) | 48.084–3.545 (3.774–3.545) | 48.930–3.307 (3.659–3.307) | 49.307–3.175 (3.788–3.175) |
| Observed reflections | 94,080 (4,377) | 165,652 (9,501) | 313,836 (16,833) | 896,835 (43,613) | 455,867 (22,410) | 948,639 (46,128) | 3,470,467 (207,156) | 353,906 (17,408) | 314,373 (18,872) |
| Unique reflections | 13,616 (681) | 14,606 (859) | 23,475 (1,201) | 67,576 (3,379) | 18,249 (913) | 24,304 (1,215) | 21,897 (1,288) | 17,940 (901) | 11,177 (559) |
| *$R_{merge}$ | 0.125 (1.253) | 0.166 (1.797) | 0.153 (1.979) | 0.090 (2.855) | 0.093 (3.060) | 0.188 (6.557) | 0.282 (4.684) | 0.115 (1.987) | 0.367 (3.891) |
| †$R_{pim}$ | 0.051 (0.528) | 0.051 (0.552) | 0.043 (0.545) | 0.026 (0.822) | 0.019 (0.631) | 0.030 (1.074) | 0.023 (0.368) | 0.026 (0.455) | 0.069 (0.670) |
| $CC_{1/2}$ | 0.997 (0.445) | 0.995 (0.602) | 0.999 (0.619) | 0.998 (0.429) | 0.999 (0.440) | 0.999 (0.363) | 1.000 (0.331) | 0.999 (0.012) | 0.999 (0.726) |
| ‡$<I/\sigma I>$ | 9.0 (1.5) | 11.5 (1.7) | 13.1 (1.5) | 12.9 (1.0) | 18.3 (1.3) | 16.9 (0.8) | 27.3 (2.3) | 11.1 (1.8) | 9.3 (1.3) |
| Overall completeness spherical/elliptical | 51.8/89.9 (9.9/61.5) | 72.4/91.3 (16.3/57.4) | 81.1/91.7 (21.3/50.7) | 71.1/94.1 (13.8/65.1) | 76.8/93.9 (19.1/61.0) | 69.4/94.9 (15.0/64.2) | 88.5/95.4 (31.2/53.3) | 47.2/93.4 (9.2/82.3) | 88.7/28.9 (49.1/3.6) |
| Multiplicity | 6.9 (6.4) | 11.3 (11.1) | 13.4 (14.0) | 13.3 (12.9) | 25.0 (24.5) | 39.0/38.0 | 158.5 (160.8) | 19.7 (19.3) | 28.1 (33.8) |
| Wilson B-factor (Å²) | 26.91 | 38.32 | 18.39 | 23.28 | 63.11 | 48.24 | 150.24 | 145.19 | 97.08 |
| No. reflections | 13,592 (398) | 14,580 (438) | 23,131 (438) | 67,483 (68) | 18,244 (377) | 24,283 (185) | 21,715 (432) | 17,917 (462) | 10,621 (108) |
| §$R_{work}$/¶$R_{free}$ | 0.2005/ 0.2396 | 0.2018/ 0.2494 | 0.2102/ 0.2470 | 0.1989/ 0.2254 | 0.2379/ 0.2645 | 0.2280/ 0.2549 | 0.2824/ 0.2954 | 0.2292/ 0.2630 | 0.2989/ 0.3110 |
| No. of non-hydrogen atoms | 2287 | 2236 | 2341 | 4664 | 2101 | 2229 | 8379 | 4253 | 10,579 |
| Protein | 2242 | 2173 | 2177 | 4278 | 2101 | 2200 | 8379 | 4253 | 10,572 |
| Ligands | 3 | 0 | 0 | 0 | 0 | 5 | 0 | 0 | 7 |
| Solvent | 42 | 63 | 164 | 386 | 0 | 24 | 0 | 0 | 0 |
| Protein residues | 284 | 274 | 275 | 500 | 255 | 272 | 1031 | 509 | 1330 |
| **Ramachandran statistics: favored/ allowed (%) | 97.12/2.88 | 99.62/0.38 | 99.63/0.37 | 99.19/0.81 | 98.41/1.59 | 96.59/3.03 | 95.01/4.09 | 96.61/2.14 | 96.74/3.11 |
| Clashscore | 1.55 | 0.69 | 1.84 | 0.94 | 1.20 | 4.33 | 11.36 | 11.12 | 7.43 |
| Overall B-factor (Å²) | 36.15 | 45.76 | 25.40 | 31.92 | 72.62 | 64.16 | 179.72 | 156.98 | 98.05 |
| RMS deviations in | | | | | | | | | |
| Bonds (Å) | 0.003 | 0.002 | 0.004 | 0.005 | 0.002 | 0.003 | 0.004 | 0.003 | 0.005 |
| Angles (°) | 0.51 | 0.47 | 0.64 | 0.70 | 0.47 | 0.56 | 0.82 | 0.67 | 0.89 |

*Table 1 continued on next page*

*Table 1 continued*

| | RanBP9 CTLH-CRA | RanBP10 CTLH-CRA | R10_GLFF CTLH-CRA | Maea CTLH-CRA | Twa1 CTLH-CRA | Mkln-R9 CTLH-CRA | R10_GLFF-Maea CTLH-CRA | Twa_14-Maea CTLH-CRA | RanBP10-Twa1 |
|---|---|---|---|---|---|---|---|---|---|

Numbers in parentheses refer to the respective highest resolution data shell in the dataset.

*$R_{sym} = \Sigma_{hkl} \Sigma_i | I_i - <I> | / \Sigma_{hkl} \Sigma_i I_i$ where $I_i$ is the $i^{th}$ measurement and $<I>$ is the weighted mean of all measurements of I.

†$R_{pim} = \Sigma_{hkl} 1/(N–1)^{1/2} \Sigma_i | I_i(hkl) – I (hkl) | / \Sigma_{hkl} \Sigma_i I(hkl)$, where N is the redundancy of the data and I (hkl) the average intensity.

‡$<I / \sigma I>$ indicates the average of the intensity divided by its standard deviation.

§$R_{work} = \Sigma_{hkl} ||F_o| – |F_c|| / \Sigma_{hkl}|F_o|$, where $F_o$ and $F_c$ are the observed and calculated structure factor amplitudes, respectively.

¶$R_{free}$ same as R for 5% of the data randomly omitted from the refinement. The number of reflections includes the $R_{free}$ subset.

**Ramachandran statistics were calculated using MolProbity in Phenix.

# Additional files

## Supplementary files
MDAR checklist

## Data availability
The models of the CTLH complex subunits are deposited in the Protein Data Bank with the following accession codes: 9SNE, 9SNF, 9SNG, 9SNH, 9SNI, 9SNV, 9SOH, 9SOI, and 9SOC. All other materials, e.g. primers or plasmids, are available upon request.

The following datasets were generated:

| Author(s) | Year | Dataset title | Dataset URL | Database and Identifier |
|---|---|---|---|---|
| van gen Hassend PM, Schindelin H | 2026 | CTLH-CRA domain of murine RanBP9 | https://doi.org/10.2210/pdb9sne/pdb | Worldwide Protein Data Bank, 10.2210/pdb9sne/pdb |
| van gen Hassend PM, Schindelin H | 2026 | CTLH-CRA domain of murine RanBP10 | https://doi.org/10.2210/pdb9SNF/pdb | Worldwide Protein Data Bank, 10.2210/pdb9SNF/pdb |
| van gen Hassend PM, Schindelin H | 2026 | CTLH-CRA domain of murine RanBP10 mutant Q519G, F543L, Y548F and V556F | https://doi.org/10.2210/pdb9SNG/pdb | Worldwide Protein Data Bank, 10.2210/pdb9SNG/pdb |
| van gen Hassend PM, Schindelin H | 2026 | CTLH-CRA domain of murine Maea | https://doi.org/10.2210/pdb9SNH/pdb | Worldwide Protein Data Bank, 10.2210/pdb9SNH/pdb |
| van gen Hassend PM, Schindelin H | 2026 | CTLH-CRA domain of murine Twa1 | https://doi.org/10.2210/pdb9SNI/pdb | Worldwide Protein Data Bank, 10.2210/pdb9SNI/pdb |
| van gen Hassend PM, Schindelin H | 2026 | CTLH-CRA domains of the RanBP9-muskelin complex | https://doi.org/10.2210/pdb9SNV/pdb | Worldwide Protein Data Bank, 10.2210/pdb9SNV/pdb |
| van gen Hassend PM, Schindelin H | 2026 | CTLH-CRA domains of the Maea-RanBP10 mutant (Q519G, F543L, Y548F and V556F) complex | https://doi.org/10.2210/pdb9SOH/pdb | Worldwide Protein Data Bank, 10.2210/pdb9SOH/pdb |
| van gen Hassend PM, Schindelin H | 2026 | CTLH-CRA domains of Maea-Twa1 mutant (A125G, Q126R, T127del, Q128E, A130Q, M143L, E144Q, L147F, A148S, F152Y and F160V) complex | https://doi.org/10.2210/pdb9SOI/pdb | Worldwide Protein Data Bank, 10.2210/pdb9SOI/pdb |
| van gen Hassend PM, Schindelin H | 2026 | Murine RanBP10-Twa1 complex | https://doi.org/10.2210/pdb9SOC/pdb | Worldwide Protein Data Bank, 10.2210/pdb9SOC/pdb |

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
