## [Editor Report · eLife Assessment]

This structural biology study provides insights into the assembly of the GID/CTLH E3 ligase complex. The multi-subunit complex forms unique, ring-shaped assemblies and the findings presented here describe a "specificity code" that regulates formation of subunit interfaces. The data supporting the conclusions are **convincing**, both in thoroughness and rigor. This study will be **valuable** to biochemists, structural biologists, and could lay foundation for novel designed protein assemblies.

---

## [Referee Report · Reviewer #1 (Public review)]

Summary:

GID/CTLH-type RING ligases are huge multi-protein complexes that play an important role in protein ubiquitylation. The subunits of its core complex are distinct and form a defined structural arrangement, but there can be variations in subunit composition, such as exchange of RanBP9 and RanBP10. In this study, van gen Hassend and Schindelin provide new crystal structures of (parts of) key subunits and use those structures to elucidate the molecular details of the pairwise binding between those subunits. They identify key residues that mediate binding partner specificity. Using in vitro binding assays with purified protein, they show that altering those residues can switch specificity to a different binding partner.

Strengths:

This is a technically demanding study that sheds light on an interesting structural biology problem in residue-level detail. The combination of crystallization, structural modeling and binding assays with purified mutant proteins is elegant and, in my eyes, convincing.

Weaknesses:

This study has no major weaknesses.

It will be very interesting to see follow-up studies that use the mutants generated here to dive deeper into the biology of RING ligases, or design new mutants of multi-subunit complexes with an analogous methodology.

---

## [Referee Report · Reviewer #2 (Public review)]

Summary:

This is a very interesting study focusing on a remarkable oligomerization domain, the LisH-CTLH-CRA module. The module is found in a diverse set of proteins across evolution. The present manuscript focuses on the extraordinary elaboration of this domain in GID/CTLH RING E3 ubiquitin ligases, which assemble into a gigantic, highly ordered, oval-shaped megadalton complex with strict subunit specificity. The arrangement of LisH-CTLH-CRA modules from several distinct subunits is required to form the oval on the outside of the assembly, allowing functional entities to recruit and modify substrates in the center. Although previous structures had shown that data revealed that CTLH-CRA dimerization interfaces share a conserved helical architecture, the molecular rules that govern subunit pairing have not been explored. This was a daunting task in protein biochemistry that was achieved in the present study, which defines this "assembly specificity code" at the structural and residue-specific level.

The authors used X-ray crystallography to solve high-resolution structures of mammalian CTLH-CRA domains, including RANBP9, RANBP10, TWA1, MAEA, and the heterodimeric complex between RANBP9 and MKLN. They further examined and characterized assemblies by quantitative methods (ITC and SEC-MALS) and qualitatively using nondenaturing gels. Some of their ITC measurements were particularly clever, and involved competitive titrations, and titrations of varying partners depending on protein behavior. The experiments allowed the authors to discover that affinities for interactions between partners is exceptionally tight, in the pM-nM range, and to distill the basis for specificity while also inferring that additional interactions beyond the LisH-CTLH-CRA modules likely also contribute to stability. Beyond discovering how the native pairings are achieved, the authors were able to use this new structural knowledge to reengineer interfaces to achieve different preferred partnerings.

Strengths:

Nearly everything about this work is exceptionally strong.

-The question is interesting for the native complexes, and even beyond that has potential implications for design of novel molecular machines.

-The experimental data and analyses are quantitative, rigorous, and thorough.

-The paper is a great read - scholarly and really interesting.

-The figures are exceptional in every possible way. They present very complex and intricate interactions with exquisite clarity. The authors are to be commended for outstanding use of color and color-coding throughout the study, including in cartoons to help track what was studied in what experiments. And the figures are also outstanding aesthetically.

Weaknesses:

There are no major weaknesses of note, and in the revision the authors addressed my minor suggestions for the text.

---

## [Referee Report · Reviewer #3 (Public review)]

Summary:

Protein complexes, like the GID/CTLH-type E3 ligase, adopt a complex three-dimensional structure, which is of functional importance. Several domains are known to be involved in shaping the complexes. Structural information based on cryo-EM is available, but its resolution does not always provide detailed information on protein-protein interactions. The work by van gen Hassend and Schindelin provides additional structural data based on crystal structures.

Strengths:

The work is solid and very carefully performed. It provides high-resolution insights into the domain architecture, which helps to understand the protein-protein interactions on a detailed molecular level. They also include mutant data and can thereby draw conclusions on the specificity of the domain interactions. These data are probably very helpful for others who work on a functional level with protein complexes containing these domains.

Weaknesses:

The manuscript contains a lot of useful, very detailed information. This information is likely very helpful to investigate functional and regulatory aspects of the protein complexes, whose assembly relies on the LisH-CTLH-CRA modules. However, this goes beyond the scope of this manuscript.

Comments on revisions:

I am fine with the revised version of the manuscript.

---

## [Author Response]

The following is the authors’ response to the original reviews.

**Public Reviews:**

**Reviewer #1 (Public review):**
Summary:GID/CTLH-type RING ligases are huge multi-protein complexes that play an important role in protein ubiquitylation. The subunits of its core complex are distinct and form a defined structural arrangement, but there can be variations in subunit composition, such as exchange of RanBP9 and RanBP10. In this study, van gen Hassend and Schindelin provide new crystal structures of (parts of) key subunits and use those structures to elucidate the molecular details of the pairwise binding between those subunits. They identify key residues that mediate binding partner specificity. Using in vitro binding assays with purified protein, they show that altering those residues can switch specificity to a different binding partner.Strengths:This is a technically demanding study that sheds light on an interesting structural biology problem in residue-level detail. The combination of crystallization, structural modeling, and binding assays with purified mutant proteins is elegant and, in my eyes, convincing.Weaknesses:I mainly have some suggestions for further clarification, especially for a broad audience beyond the structural biology community.

We thank the reviewer for the careful evaluation of our manuscript and for the positive and encouraging assessment of our work. We also thank the reviewer for the constructive suggestions to improve clarity for a broader audience and have revised the manuscript accordingly.

(1) The authors establish what they call an 'engineering toolkit' for the controlled assembly of alternative compositions of the GID complex. The mutagenesis results are great for the specific questions asked in this manuscript. It would be great if they could elaborate on the more general significance of this 'toolkit' - is there anything from a technical point of view that can be generalized? Is there a biological interest in altering the ring composition for functional studies?

We thank the reviewer for raising this important point. Beyond addressing the specific pairwise assembly mechanisms analyzed in this study, we agree that the broader significance of this engineering toolkit warrants further discussion. The residue-level understanding of CTLH-CRA interfaces not only explains assembly specificity but also enables rational manipulation of ring composition in a controlled manner. We have therefore expanded the end of the discussion section to outline generalizable strategies for CRA-interface disruption and to highlight potential biological applications of altering ring composition for functional studies.

(2) Along the same lines, the mutagenesis required to rewire Twa1 binding was very complex (8 mutations). While this is impressive work, the 'big picture conclusion' from this part is not as clear as for the simpler RanBP9/10. It would be great if the authors could provide more context as to what this is useful for (e.g., potential for in vivo or in vitro functional studies, maybe even with clinical significance?)

We thank the reviewer for this important comment and agree that the broader implications of the more complex Twa1 rewiring were not sufficiently emphasized in the original manuscript. Through the competition ITC experiments (Figure 5), we aimed to demonstrate a concrete application of the Twa1. At the same time, we recognize that additional use cases are conceivable. To address this point, we have expanded the discussion section to clarify the conceptual significance of Twa1 rewiring and briefly outline further potential applications of controlled interface manipulation. These additions aim to better contextualize the broader relevance of this approach beyond the specific mechanistic questions addressed in this study.

(3) For many new crystal structures, the authors used truncated, fused, or otherwise modified versions of the proteins for technical reasons. It would be helpful if the authors could provide reasoning why those modifications are unlikely to change the conclusions of those experiments compared to the full-length proteins (which are challenging to work with for technical reasons). For instance, could the authors use folding prediction (AlphaFold) that incorporates information of their resolved structures and predicts the impact of the omitted parts of the proteins? The authors used AlphaFold for some aspects of the study, which could be expanded.

We agree with the reviewer that the transferability of the domain constructs to the corresponding full-length proteins is an important consideration. In the original version of the manuscript, we addressed this point by fitting the experimentally determined CTLH-CRA domain structures of muskelin and RanBP9 into the cryo-EM maps of the full-length complexes (Figure 5D), demonstrating that the applied truncations and fusion strategies are compatible with the architecture observed in the intact assembly. Following the reviewer’s suggestion, we have further strengthened this analysis by adding a new Supplementary Figure 1. In this figure, the experimentally determined CTLH-CRA domain structures are superposed with full-length AlphaFold predictions. This comparison shows that removal of flexible linker regions, such as those between the CTLH and CRA motifs or at terminal segments, does not alter the overall fold or the binding interfaces of the domains. Together, these analyses support the conclusion that the domain constructs faithfully represent the structural and interaction properties of the full-length proteins.

**Reviewer #2 (Public review):**
Summary:This is a very interesting study focusing on a remarkable oligomerization domain, the LisH-CTLH-CRA module. The module is found in a diverse set of proteins across evolution. The present manuscript focuses on the extraordinary elaboration of this domain in GID/CTLH RING E3 ubiquitin ligases, which assemble into a gigantic, highly ordered, oval-shaped megadalton complex with strict subunit specificity. The arrangement of LisH-CTLHCRA modules from several distinct subunits is required to form the oval on the outside of the assembly, allowing functional entities to recruit and modify substrates in the center. Although previous structures had shown that data revealed that CTLH-CRA dimerization interfaces share a conserved helical architecture, the molecular rules that govern subunit pairing have not been explored. This was a daunting task in protein biochemistry that was achieved in the present study, which defines this "assembly specificity code" at the structural and residue-specific level.The authors used X-ray crystallography to solve high-resolution structures of mammalian CTLH-CRA domains, including RANBP9, RANBP10, TWA1, MAEA, and the heterodimeric complex between RANBP9 and MKLN. They further examined and characterized assemblies by quantitative methods (ITC and SEC-MALS) and qualitatively using nondenaturing gels. Some of their ITC measurements were particularly clever and involved competitive titrations and titrations of varying partners depending on protein behavior. The experiments allowed the authors to discover that affinities for interactions between partners is exceptionally tight, in the pM-nM range, and to distill the basis for specificity while also inferring that additional interactions beyond the LisH-CTLH-CRA modules likely also contribute to stability. Beyond discovering how the native pairings are achieved, the authors were able to use this new structural knowledge to reengineer interfaces to achieve different preferred partnerings.Strengths:Nearly everything about this work is exceptionally strong.(1) The question is interesting for the native complexes, and even beyond that, has potential implications for the design of novel molecular machines.(2) The experimental data and analyses are quantitative, rigorous, and thorough.(3) The paper is a great read - scholarly and really interesting.(4) The figures are exceptional in every possible way. They present very complex and intricate interactions with exquisite clarity. The authors are to be commended for outstanding use of color and color-coding throughout the study, including in cartoons to help track what was studied in what experiments. And the figures are also outstanding aesthetically.Weaknesses:There are no major weaknesses of note, but I can make a few recommendations for editing the text.

We are very grateful to the reviewer for this exceptionally positive and thoughtful assessment of our work. We sincerely appreciate the recognition of both the conceptual scope and the technical depth of the study. We are particularly encouraged by the reviewer’s comments regarding the clarity and presentation of the figures. Considerable effort went into ensuring that the structural and biochemical complexity of the CTLH assemblies could be conveyed in a clear and accessible manner, and we are grateful that this was appreciated. We thank the reviewer for the constructive recommendations for textual improvements.

**Reviewer #3 (Public review):**
Summary:Protein complexes, like the GID/CTLH-type E3 ligase, adopt a complex three-dimensional structure, which is of functional importance. Several domains are known to be involved in shaping the complexes. Structural information based on cryo-EM is available, but its resolution does not always provide detailed information on protein-protein interactions. The work by van gen Hassend and Schindelin provides additional structural data based on crystal structures.Strengths:The work is solid and very carefully performed. It provides high-resolution insights into the domain architecture, which helps to understand the protein-protein interactions on a detailed molecular level. They also include mutant data and can thereby draw conclusions on the specificity of the domain interactions. These data are probably very helpful for others who work on a functional level with protein complexes containing these domains.Weaknesses:The manuscript contains a lot of useful, very detailed information. This information is likely very helpful to investigate functional and regulatory aspects of the protein complexes, whose assembly relies on the LisH-CTLHCRA modules. However, this goes beyond the scope of this manuscript.

We thank the reviewer for the detailed review of our manuscript and for the constructive and positive remarks. We greatly appreciate the recognition of the high-resolution structural insights and the value of combining crystallographic data with mutational analyses to elucidate domain-specific interactions. We are also grateful for the acknowledgment that these findings may serve as a useful resource for future functional and regulatory studies of LisH-CTLH-CRA-containing complexes. While such aspects extend beyond the immediate scope of the present study, we hope that the structural framework provided here will facilitate and inspire future investigations addressing these questions.

**Recommendations for the authors:**

**Reviewer #2 (Recommendations for the authors):**
(1) For the ITC measurements that are less accurate, the authors may want to represent that in the figures with an approximate sign.

We thank the reviewer for this helpful suggestion. After consideration, we decided not to introduce an approximate sign in the main figures, as this would be inconsistent with the graphical conventions used throughout the manuscript (there is also no equal sign). Since the associated errors are reported directly alongside each K_D_ value, we believe that the precision of the measurements is sufficiently conveyed. However, we agree that explicitly marking estimated values can be appropriate in specific cases. We have therefore added approximate signs in Supplementary Fig. 5 for the K_D_ estimation of self-association.

(2) The names of the proteins are from mammals and should probably be capitalized.

We agree that capitalization is generally appropriate for mammalian protein names. In particular, for proteins such as Rmnd5a, which is identical in sequence between mouse and human, the use of human-style nomenclature would indeed be fully justified. Originally, we chose the current nomenclature to distinguish the proteins studied here from strictly human versions, as most constructs are derived from mouse and one (muskelin) from rat. This approach also avoids inconsistencies between the mouse and rat proteins within the manuscript and maintains alignment with the nomenclature used in our previous publications. For the sake of consistency and continuity, we have therefore retained the original formatting throughout the manuscript.

(3) For the sequence alignments, it would be good to specify in the legend which organisms these are from, and where the differences are in mouse and rat proteins used in the study, and the human proteins.

We appreciate this constructive suggestion. We have revised the sequence alignment legends to clearly specify the organism of origin for each sequence included in the analysis. In addition, we have added a new Supplementary Figure 1 presenting the AlphaFold predictions of the mouse proteins and rat muskelin used in this study. Within these models, sequence differences relative to the human proteins are indicated, and variations within the CTLH-CRA domains are explicitly annotated. These additions clarify how the constructs analyzed here relate to their human counterparts.

(4) A few points about the referencing:(a) It was reference 27 that first described the dual-sided interactions where the CRA domain weaves back and forth such that CTLH-CRAN and LisH-CRAC mediate the contacts on the two sides. This should be cited.

We fully agree and added the reference accordingly.

(b) To this reviewer's knowledge, it was references 13 and 9 that resolved the daisy-chain of helical LisH-CTLHCRA interactions around the oval helical structures.

We agree with the reviewer that references 13 and 9 resolved the helical LisH-CTLH-CRA daisy-chain arrangement around the oval structure. Reference 13 was already cited in the original manuscript, and we have now added reference 9 to appropriately acknowledge this contribution. We have retained reference 14, although it did not resolve the helical daisy-chain architecture, as it described a related oval assembly of CTLH complex components that remains relevant in the structural context discussed.

(c) A cryo-EM map with RANBP10 was shown at low resolution in reference 8.

We agree with the reviewer that a low-resolution cryo-EM map including RANBP10 was reported in reference 8. Our original wording was not sufficiently precise and may have given the impression that RANBP10 had not been characterized. Our intention was to convey that, although cryo-EM maps exist, detailed atomic-level information on subunit interfaces was lacking. We have revised the paragraph accordingly to clarify this point and now cite reference 8 explicitly in this context.

(d) The Discussion requires referencing.

We agree with the reviewer that additional referencing improves the clarity and contextualization of the Discussion. We have revised the Discussion section accordingly and added appropriate references to support the statements made.